# Adapt then Unlearn: Exploring Parameter Space Semantics for Unlearning in Generative Adversarial Networks

**Piyush Tiwary**                                                        *piyushtiwary@iisc.ac.in*
*Department of Electrical Communication Engineering*
*Indian Institute of Science*

**Atri Guha**                                                          *atri_2001ee08@iitp.ac.in*
*Department of Electrical Engineering*
*Indian Institute of Technology Patna*

**Subhodip Panda**                                                        *subhodipp@iisc.ac.in*
*Department of Electrical Communication Engineering*
*Indian Institute of Science*

**Prathosh A.P.**                                                          *prathosh@iisc.ac.in*
*Department of Electrical Communication Engineering*
*Indian Institute of Science*

**Reviewed on OpenReview:** *https://openreview.net/forum?id=jAHEBivObO*

## Abstract

Owing to the growing concerns about privacy and regulatory compliance, it is desirable to regulate the output of generative models. To that end, the objective of this work is to prevent the generation of outputs containing undesired features from a pre-trained Generative Adversarial Network (GAN) where the underlying training data set is inaccessible. Our approach is inspired by the observation that the parameter space of GANs exhibits meaningful directions that can be leveraged to suppress specific undesired features. However, such directions usually result in the degradation of the quality of generated samples. Our proposed two-stage method, known as '**Adapt-then-Unlearn**,' excels at unlearning such undesirable features while also maintaining the quality of generated samples. In the initial stage, we adapt a pre-trained GAN on a set of negative samples (containing undesired features) provided by the user. Subsequently, we train the original pre-trained GAN using positive samples, along with a repulsion regularizer. This regularizer encourages the learned model parameters to move away from the parameters of the adapted model (first stage) while not degrading the generation quality. We provide theoretical insights into the proposed method. To the best of our knowledge, our approach stands as the first method addressing unlearning within the realm of high-fidelity GANs (such as StyleGAN). We validate the effectiveness of our method through comprehensive experiments, encompassing both class-level unlearning on the MNIST and AFHQ dataset and feature-level unlearning tasks on the CelebA-HQ dataset. Our code and implementation is available at: https://github.com/atriguha/Adapt_Unlearn.

## 1 Introduction

### 1.1 Unlearning

Recent advancements in deep generative models such as Generative Adversarial Networks (GANs) (Goodfellow et al., 2014; Arjovsky et al., 2017; Karras et al., 2018b;a; 2020) and Diffusion models (Ho et al., 2020;

Song & Ermon, 2019; Song et al., 2021) have showcased remarkable performance in diverse tasks, from generating high-fidelity images (Karras et al., 2018a; 2020; 2021) to text-to-image translations (Ramesh et al., 2021; 2022; Rombach et al., 2022). Consequently, these models find application in various fields, including but not limited to medical imaging (Celard et al., 2023; Varoquaux & Cheplygina, 2022; Tiwary et al., 2024a), remote sensing (Ball et al., 2017; Adegun et al., 2023), hyperspectral imagery (Jia et al., 2021; Wang et al., 2023), and many others (Choudhary et al., 2022; Yang & Xu, 2021; Liu et al., 2021; Tiwary et al., 2024b). However, the extensive incorporation of data with possible undesired features or inherent biases cause these models to generate violent, racial, or explicit content which poses significant concerns (Tommasi et al., 2017). Thus, these models are subject to regulatory measures (Voigt & dem Bussche, 2017; Goldman, 2020). Identifying and eliminating these undesired features from the model's knowledge representation poses a challenging task. The framework of Machine Unlearning (Xu et al., 2020; Nguyen et al., 2022b) tries to solve this problem by removing specific training data points containing undesired feature from the pre-trained model. Specifically, machine unlearning refers to the task of forgetting the learned information (Sekhari et al., 2021; Ma et al., 2022; Ye et al., 2022; Cao & Yang, 2015; Golatkar et al., 2021; 2020a; Ginart et al., 2019; Golatkar et al., 2020b), or erasing the influence of specific data subset of the training dataset from a trained model in response to a user request (Wu et al., 2020a; Guo et al., 2020; Graves et al., 2021; Wu et al., 2022; 2020b; Chourasia & Shah, 2023).

The task of unlearning can be challenging because we aim to '*unlearn*' a specific undesired feature without negatively impacting the previously acquired knowledge. In other words, unlearning could lead to Catastrophic Forgetting (Ginart et al., 2019; Nguyen et al., 2022a; Golatkar et al., 2020b) which would significantly deteriorate the performance of the model. Further, the level of difficulty faced in the process of unlearning may vary depending on the specific features of the data that one is required to unlearn. For example, unlearning a particular class (e.g. class of digit '9' in MNIST) could be relatively easier than unlearning a more subtle feature (e.g. beard feature in CelebA). This is because the classes in MNIST are quite distinct and don't necessarily share correlated features. Whereas, in the CelebA (Liu et al., 2015) dataset, the feature of having a beard is closely linked to the concept of gender. So, unlearning this subtle feature while retaining other correlated features such as gender, poses an increasingly difficult challenge. It is important to mention that re-training the model from scratch *without* the undesired input data is not feasible in this setting due unavailability of the training dataset.

## 1.2 Motivation and Contribution

In this work, we try to solve the problem of unlearning undesired feature in pre-trained generative adversarial networks (GANs) *without* having access to the training data used for pre-training the GAN. We operate under the feedback-based unlearning framework, where we start with a pre-trained GAN. A user is given a set of generated samples from this GAN. The user chooses a subset of generated samples and identifies them as 'undesirable' (negative samples). The feedback-based approach is similar to RLHF in LLMs or human-in-loop settings in general (Ziegler et al., 2019; Christiano et al., 2017; Lambert et al., 2022). The objective of the process of unlearning is to prevent the generation of undesirable characteristics, as identified by the user. We propose to unlearn the undesired features by following a two-step approach. In the first step, we adapt the pre-trained generator to the undesired features by using the samples marked as undesired by the user (negative samples). This ensures that the '*adapted*' generator exclusively generates samples that possess the undesired features. In the next step, we unlearn the undesired features from the original GAN by using the samples that weren't marked as undesired by the user (positive samples). While unlearning, we add a *repulsion* loss that encourages the parameters of the generator to stay away from the parameters of the adapted generator (obtained from first step) while also making sure that the quality of generated samples does not deteriorate. We provide theoretical justification for the proposed method by using a bayesian framework. Particularly, we show that the proposed method leads to contrastive-divergence kind of objective desired for unlearning. We call the proposed two-stage process '**Adapt-then-Unlearn**'. An overview of the proposed method is shown in figure 1 (a).

Our approach hinges in realizing interpretable and meaningful directions within the parameter space of a pre-trained GAN generator, as discussed in (Cherepkov et al., 2021). In particular, the first stage of the proposed method leads to adapted parameters that exclusively generate negative samples. While the

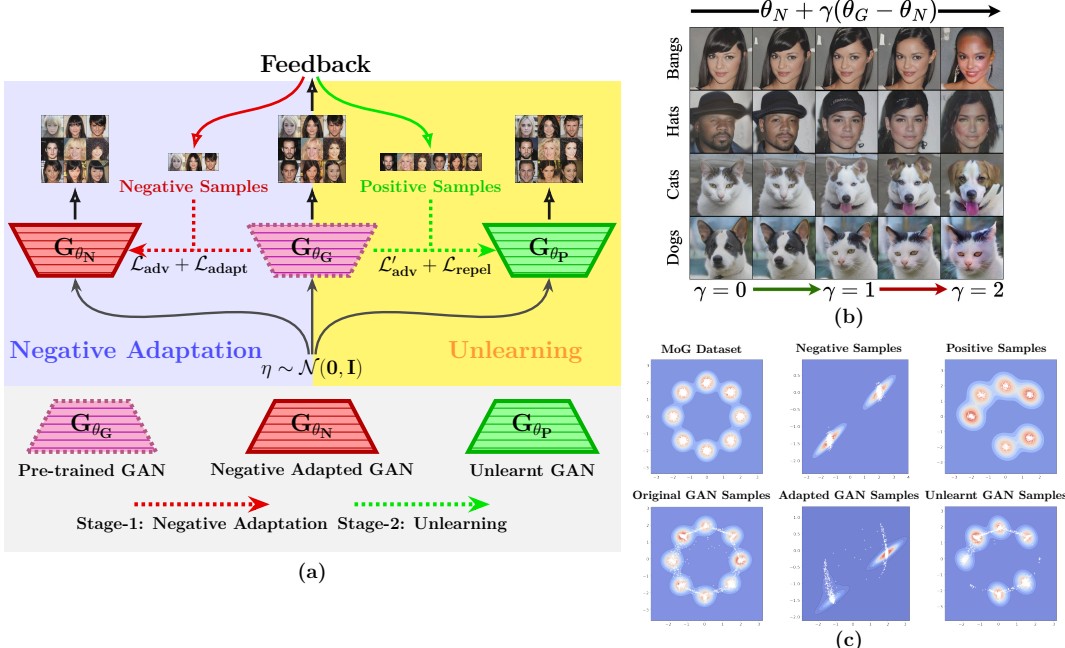

Figure 1: (a) Block diagram of the proposed method: Stage-1: Negative Adaptation of the GAN to negative samples received from user feedback and Stage-2: Unlearning of the original GAN using the positive samples with a repulsion loss. (b) Illustrating linear interpolation and extrapolation in parameter space for unlearning undesired features. We observe that in the extrapolation region, undesired features are suppressed, but the quality of generated samples deteriorates. (c) An example of results obtained using our method on Mixture of Gaussian (MoG) dataset, where we unlearn two centers provided in negative samples.

parameters of the original pre-trained generator generate both positive as well as negative samples. Hence, the difference between the parameters of adapted generator and the paramters of original generator can be interpreted as the direction in parameter space that leads to a decrease in the generation of negative samples. Given this, it is sensible to move away from the original parameters in this direction to further reduce the generation of negative samples. This observation is shown in figure 1 (b). However, it's worth noting that such extrapolation doesn't ensure the preservation of other image features' quality. In fact, deviations too far from the original parameters may hamper the smoothness of the latent space, potentially leading to a deterioration in the overall generation quality (see last columns of figure 1 (b)). Inspired by this observation, during unlearning stage, we propose to train the generator using adversarial loss while encouraging the generator parameters to be away from the parameters of the adapted generator by employing a repulsion regularization.

We provide a visual illustration of the proposed method on Mixture of Gaussian (MoG) dataset with eight centers in figure 1 (c). The first column shows the original training dataset and the samples generated by the pre-trained GAN. The second column shows the negative samples provided during feedback and the samples generated by the adapted generator. We can see that the adapted generator exclusively generates samples from the negative modes of MoG. Lastly, in the third column, we see the positive samples and the samples generated after unlearning the negative modes. We clearly observe that after unlearning (via the proposed method), the generator unlearns the negative modes and generates samples from the rest of the modes. This gives a proof-of-concept for the proposed method.

We summarize our contribution as follows:

- We introduce a two-stage approach for machine unlearning in GANs. In the first stage, our method adapts the pre-trained GAN to the negative samples. In the second stage, we train the GAN using

a repulsion loss, ensuring that the generator's parameters diverge from those of the adapted GAN in stage 1.

- By design, our method can operate in practical few-shot settings where the user provides a very small amount of negative samples.

- We provide theoretical justification for the proposed method by showing that the proposed regularization leads to contrastive-divergence kind of objectives appropriate for unlearning.

- The proposed method is thoroughly tested on multiple datasets, considering various types of unlearning scenarios such as class-level unlearning and feature-level unlearning. Throughout these tests, we empirically observe that the quality of the generated samples is not compromised.

## 2 Related Work

### 2.1 Machine Unlearning

Unlearning can be naively done by removing the unwanted data subset from the training dataset and then retraining the model from scratch. However, retraining is computationally costly and becomes impossible if the unlearning request comes recursively for single data points. The task of recursively *'unlearning'* i.e. removing information of a single data point in an online manner (also known as decremental learning) for the SVM algorithm was introduced in (Cauwenberghs & Poggio, 2000). However, when multiple data points are added or removed, these algorithms become slow because they need to be applied to each data point individually. To address this, (Karasuyama & Takeuchi, 2009) introduced a newer type of SVM training algorithm that can efficiently update an SVM model when multiple data points are added or removed simultaneously. Later, inspired by the problem of protecting user privacy (Cao & Yang, 2015) developed efficient ways to delete data from certain statistical query algorithms and coined the term "machine unlearning". The works of (Ginart et al., 2019) extended the idea of unlearning to more complicated algorithms such k-means clustering and also proposed the first definition of effective data deletion that can be applied to randomized algorithms, in terms of statistical indistinguishability. Depending upon this statistical indistinguishability criteria machine unlearning processes are widely classified into exact unlearning (Ginart et al., 2019; Brophy & Lowd, 2021) and approximate unlearning methods (Neel et al., 2021; Nguyen et al., 2020). The goal of exact unlearning is to exactly match the parameter distributions of the unlearned model and the retrained model where as, in approximate unlearning, the distributions of the unlearned and retrained model's parameters are close to some small multiplicative and additive terms (Neel et al., 2021). To extend the idea of unlearning or efficient data deletion for non-convex models such as deep neural networks (Golatkar et al., 2020b) proposed a scrubbing mechanism for approximate unlearning in deep neural networks. A more efficient method of unlearning in deep networks is proposed by (Goel et al., 2022) where the initial layers of deep networks are frozen while the last few layers are finetuned on the filtered dataset. Further to achieve the goal of exact unlearning (Jia et al., 2023) exploit the model sparsification technique via weight pruning. Even though all of these methods achieve unlearning in supervised deep networks, the generalization of these methods for state-of-the-art high-fidelity GANs is unexplored.

Few methods like cascaded unlearning (Sun et al., 2023) and data redaction (Kong & Chaudhuri, 2023) try to prevent generation of undesired features in GANs, however, their methods operate primarily on very primitive DC-GAN as opposed to high-fidelity GANs like StyleGAN which is the focus of this work. While Sun et al. (2023) also show result on StyleGAN, there are several significant differences compared to the proposed method. First, there is a fundamental difference in the unlearning setting between Sun et al. (2023) and our method. To reduce the generation of undesired samples, Sun et al. (2023) proposes to forget undesired samples from the training dataset. Specifically, they assume access to samples from the training dataset. This is somewhat restrictive since users typically don't have access to the training data (Chundawat1 et al., 2023; Graves et al., 2021). Further, in terms of methodology, Sun et al. (2023) propose to patch the latent space of the GAN with representative samples. They suggest various strategies for generating these representative samples, such as using 'average samples' or 'other class samples' (cf. Section 4.3 of their paper). However, imposing such constraints on the latent space may lead to suboptimal

latent-space semantics, potentially harming the quality of generated images. To address this, we avoid manipulating the latent space directly. Instead, we focus on parameter-space semantics, where we identify generator parameters that produce undesired samples (Stage-1: Negative Adaptation), and then retrain the GAN to avoid these parameters (Stage-2: Unlearning Phase). Additionally, since the latent space naturally adjusts based on changes in the parameter space (as shown in figure 1 (b)), we find it sufficient to focus on parameter-space semantics alone, as it automatically handles latent-space semantics as well. To the best of our knowledge, these insights into parameter-space semantics have not been explored in the context of unlearning, making our approach novel.

## 2.2 Few-Shot Generative Domain Adaptation

The area of few-shot generative domain adaptation deals with the problem where a pre-trained generative model is adapted to a target domain using very few samples. A general strategy to do this is to fine-tune the model on target data using appropriate regularizers. Eg. Wang et al. (2018) observed that using a single pre-trained GAN for fine-tuning is good enough for adaptation. However, due to the limited amount of target data, this could lead to mode collapse, hence Noguchi & Harada (2019) proposed to fine-tune only the batch statistics of the model. Although, such a strategy can be very restrictive in practice. To overcome this issue, Wang et al. (2020) proposed to append a 'miner' network before the generator. They propose a two-stage framework, where the miner network is first trained to transform the input latent space to capture the target domain distribution then the whole pipeline is re-trained using target data. While these fine-tuning based methods give equal weightage to all the parameters of the generator, Li et al. (2020) proposed to fine-tune the parameter using Elastic Weight Consolidation (EWC). Particularly, EWC is used to penalize large changes in important parameters. This importance is quantified using Fisher-information while adapting the pre-trained GAN. Mo et al. (2020) showed that fine-tuning a GAN by freezing the lower layers of discriminator is good enough in few-shot setting. Recently, a string of work (Ojha et al., 2021; Xiao et al., 2022; Lee et al., 2021) focuses on few-shot adaptation by preserving the cross-domain correspondence. Lastly, Mondal et al. (2022) suggested an inference-time optimization approach where a they prepend a latent-learner, and the latent-learner is optimized every time a new set of images are to be generated from target domain.

As mentioned earlier, our approach involves an adaptation stage, where we adapt the pre-trained GAN to the negative samples provided by the user. In practice, the amount of negative samples provided by the user is very less hence such an adaptation falls under the category of few-shot generative domain adaptation. Hence, we make use of EWC (Li et al., 2020) for this adaptation phase (cf. Section 3.2 for details).

## 3 Proposed Methodology

### 3.1 Problem Formulation and Method Overview

Consider the generator $G_{\theta_G}$ of a pre-trained GAN with parameters $\theta_G$ and an implicit generator distribution $p_G(y)$. The GAN is trained using a dataset $\mathcal{D} = \{\mathbf{x}_i\}_{i=1}^{|\mathcal{D}|}$, where $\mathbf{x}_i \overset{iid}{\sim} p_{data}(x)$. Using the feedback-based framework (Moon et al., 2023), we obtain a few negative and positive samples, marked by the user. Specifically, the user is provided with $n$ samples $\mathcal{S} = \{\mathbf{y}_i\}_{i=1}^{n}$ where $\mathbf{y}_i$ are the generated samples from the pre-trained GAN, i.e., $\mathbf{y}_i \overset{iid}{\sim} p_G(y)$. The user identifies a subset of these samples $\mathcal{S}_n = \{\mathbf{y}_i\}_{i \in s_n}$, as negative samples or samples with undesired features, and the rest of the samples $\mathcal{S}_p = \{\mathbf{y}_i\}_{i \in s_p}$ as positive samples or samples that don't possess the undesired features. Here, $s_p$ and $s_n$ are index sets such that $s_p \cup s_n = \{1, 2, \ldots, n\}$ and $s_p \cap s_n = \phi$. Formally, $\{\mathbf{y}_i\}_{i \in s_n} \overset{iid}{\sim} p_N(y)$ and $\{\mathbf{y}_i\}_{i \in s_p} \overset{iid}{\sim} p_{G \setminus N}(y)$, where, $p_N(y)$ is the implicit generator distribution on negative samples and $p_{G \setminus N}(y)$ is the implicit generator distribution after removing support of negative samples. Given this, the goal of unlearning is to learn the parameters $\theta_P$ such that the generator $G_{\theta_P}$ generates only positive samples. In other words, the parameters $\theta_P$ should lead to unlearning of the undesired features.

Our approach involves two stages: In Stage 1, we adapt the pre-trained generator $G_{\theta_G}$ on the user-marked negative samples. This step gives us the parameters $\theta_N$ such that $G_{\theta_N}$ generates only negative samples. In

Stage 2, we unlearn the undesired features by training the original generator $G_{\theta_G}$ on positive samples using the usual adversarial loss while adding an additional regularization term that makes sure that the learned parameter is far from $\theta_N$. We call this regularization term *repulsion* loss as it repels the learned parameters from $\theta_N$.

## 3.2 Stage-1: Negative Adaptation

The aim of the first stage of our method is to obtain parameter $\theta_N$ such that the generator $G_{\theta_N}$ only generates negative samples $(\mathcal{S}_n)$. However, one thing to note here is that the number of negative samples marked by the user $(|\mathcal{S}_n|)$ might be much less in number (of the order of tens or a few hundred). Directly adapting a pre-trained GAN with a much smaller amount of samples could lead to catastrophic forgetting (McClelland et al., 1995; McCloskey & Cohen, 1989). To address this issue, we employ a few-shot GAN adaptation technique, namely, Elastic Weight Consolidation (EWC) (Li et al., 2020), mainly because of its simplicity and ease of implementation. EWC-based adaptation relies on the simple observation that the 'rate of change' of weights is different for different layers. Further, this 'rate of change' is observed to be inversely proportional to the Fisher information, $F$ of the corresponding weights, which can used for penalizing changes in weights in different layers.

In our context, we want to adapt the pre-trained GAN on the negative samples. Hence, the optimal parameter $\theta_N$ for the adapted GAN can be obtained by solving the following optimization problem:

$$\theta_N, \phi_N = \arg\min_{\theta} \max_{\phi} \mathcal{L}_{adv} + \gamma \mathcal{L}_{adapt} \tag{1}$$

$$\text{where,} \quad \mathcal{L}_{adv} = \mathbb{E}_{\mathbf{x} \sim p_N(x)} \left[ \log D_\phi(\mathbf{x}) \right] + \mathbb{E}_{\mathbf{z} \sim p_Z(z)} \left[ \log(1 - D_\phi(G_\theta(\mathbf{z}))) \right] \tag{2}$$

$$\mathcal{L}_{adapt} = \lambda \sum_i F_i(\theta_i - \theta_{G,i}), F = \mathbb{E} \left[ -\frac{\partial^2}{\partial \theta_G^2} \mathcal{L}_{\theta_G}(\mathcal{S}_n) \right] \tag{3}$$

Here, $p_Z(z)$ is the standard Gaussian prior, and $\mathcal{L}_{\theta_G}(\mathcal{S}_n)$ refers to the log-likelihood function for the samples $S_n$ generated by the GAN with parameters $\theta_G$. Specifically, $\mathcal{L}_{\theta_G}(\mathcal{S}_n) = \log p_{\theta_G}(S_n)$ which is the log-likelihood of the negative samples under the generator's distribution with parameter $\theta_G$. This term can be estimated by calculating the binary cross-entropy of the discriminator's output, $D_\phi(\mathcal{S}_n)$, as shown in (Li et al., 2020). In practice, we train multiple instances of the generator to obtain multiple $\theta_N$. Specifically, given the negative samples $\mathcal{S}_n$, we adapt the pre-trained GAN $k$ times to obtain $\{\theta_N^j\}_{j=1}^k$.

## 3.3 Stage-2: Unlearning

In the second stage of our method, the actual unlearning of undesired features takes place. In particular, this stage is motivated by the observation that there exist meaningful directions in the parameter space of the generator, shown in figure 1 (b). However, such extrapolation-based schemes could lead to degradation in the quality of generated images.

Nevertheless, the above observation indicates that traversing away from $\theta_N$ helps us to erase or unlearn the undesired features. Therefore, we ask the following question: **Can we transverse in the parameter space of a generator in such a way the parameters remain far from $\theta_N$ while making sure that the quality of generated samples doesn't degrade?** To solve this problem, we make use of the positive samples $\mathcal{S}_p$ provided by the user. Particularly, we propose to train the given GAN on the positive samples while incorporating a repulsion loss component that '*repels*' or keeps the learned parameters away from $\theta_N$. Mathematically, we obtain the parameters after unlearning $\theta_P$ by solving the following optimization problem:

$$\theta_P, \phi_P = \arg\min_{\theta} \max_{\phi} \mathcal{L}'_{adv} + \gamma \mathcal{L}_{repulsion} \tag{4}$$

$$\text{where,} \quad \mathcal{L}'_{adv} = \mathbb{E}_{\mathbf{x} \sim p_{G \setminus N}(x)} \left[ \log D_\phi(\mathbf{x}) \right] + \mathbb{E}_{\mathbf{z} \sim p_Z(z)} \left[ \log(1 - D_\phi(G_\theta(\mathbf{z}))) \right] \tag{5}$$

Here, $\mathcal{L}_{repulsion}$ is the repulsion loss. The repulsion loss is chosen such that it encourages the learned parameters to be far from $\theta_N$ obtained from Stage-1. Further, $\mathcal{L}'_{adv}$ encourages the parameters to capture the desired distribution $p_{G \setminus N}(x)$. Hence, the combination of these two terms makes sure that we transverse in the parameter space maintaining the quality of generated samples while unlearning the undesired features as well.

We emphasize that $\mathcal{L}'_{adv}$ is different from $\mathcal{L}_{adv}$ used in Stage-1. Specifically, the two adversarial terms serve different purposes: (i) $\mathcal{L}_{adv}$ is utilized during the Negative Adaptation Phase (Stage-1) to adapt the original GAN to the negative samples ($\mathcal{S}_n$), and (ii) $\mathcal{L}'_{adv}$ is applied during the Unlearning Phase (Stage-2) to retrain the original GAN on positive samples ($\mathcal{S}_p$), which are the samples **not** marked as undesired.

**Note:** Our method requires users to identify or annotate negative samples, i.e., those containing undesired features. This annotation serves to adapt the GAN to negative samples during the Negative Adaptation phase and subsequently retrain it on positive samples during the Unlearning phase. While human feedback is one approach for obtaining these samples, other methods, such as curating datasets of positive and negative samples, can also be employed. However, curating such datasets can be challenging, especially when the feature, concept, or class to be unlearned is subtle or complex and is not readily annotated in standard datasets. In such scenarios, users may need to **create** a custom dataset. By contrast, our current approach leverages human feedback to annotate readily available samples generated by the GAN, reducing the need for external dataset creation. Nonetheless, if a pre-curated dataset of positive and negative samples is available, our method can be easily adapted to use it. The GAN can be trained on this dataset to obtain the negative parameters $\theta_N$, which can then be utilized in the Unlearning phase. We demonstrate few results using such approach in Appendix.



**Algorithm: Adapt-then-Unlearn**



| **Stage-1: Negative Adaptation** | **Stage-2: Unlearning** |
|---|---|
| **Required:** Pre-trained parameters ($\theta_G$, $\phi_D$), Negative samples ($\mathcal{S}_n$), Number of adapted models ($k$). | **Required:** Pre-trained parameters ($\theta_G$, $\phi_D$), Positive samples ($\mathcal{S}_p$), Adapted models ($\theta_N = \{\theta_N^j\}_{j=1}^k$). |
| **Initialize:** $j \leftarrow 0$ | **Initialize:** $\theta_P \leftarrow \theta_G$, $\phi_P \leftarrow \phi_D$ |
| 1: **while** $j \leq k$ **do**
2:     $\theta \leftarrow \theta_G$, $\phi \leftarrow \phi_D$
3:     **repeat**
4:         Sample $\mathbf{x} \sim \mathcal{S}_n$ and $\mathbf{z} \sim \mathcal{N}(0, I)$
5:         $\mathcal{L}_{adv} \leftarrow \log D_\phi(\mathbf{x}) + \log\left(1 - D_\phi(G_\theta(\mathbf{z}))\right)$
6:         $\mathcal{L}_{adapt} \leftarrow \lambda \sum_i F_i(\theta_i - \theta_{G,i})$
7:         $\theta \leftarrow \theta - \eta \nabla_\theta(\mathcal{L}_{adv} + \mathcal{L}_{adapt})$
8:     **until** convergence
9:     $\theta_N^j \leftarrow \theta$
10:    $j \leftarrow j + 1$
11: **end while** | 1: **repeat**
2:     Sample $\mathbf{x} \sim \mathcal{S}_p$ and $\mathbf{z} \sim \mathcal{N}(0, I)$
3:     $\mathcal{L}'_{adv} \leftarrow \log D_\phi(\mathbf{x}) + \log\left(1 - D_\phi(G_\theta(\mathbf{z}))\right)$
4:     Choose $\mathcal{L}_{repulsion}$ from Eq. 6
5:     $\theta \leftarrow \theta - \eta \nabla_\theta(\mathcal{L}'_{adv} + \mathcal{L}_{repulsion})$
6: **until** convergence |

## 3.4 Choice of Repulsion Loss

As mentioned above, the repulsion loss should encourage the learned parameter to traverse away from $\theta_N$ obtained from the negative adaptation stage. Here, we note that the repulsion term operates in parameter-space of the generator. There is a lineage of research work in Bayesian learning called Deep Ensembles, where multiple MAP estimates of a network are used to approximate full-data posterior (Levin et al., 1990; Hansen & Salamon, 1990; Breiman, 1996; Lakshminarayanan et al., 2017; Ovadia et al., 2019; Wilson & Izmailov, 2020; D'Angelo & Fortuin, 2021). However, if the members of an ensemble are not diverse enough, then the posterior approximation might not capture the multi-modal nature of full-data posterior. As a consequence, there are several methods proposed to increase the diversity of the members of the ensemble (Huang et al., 2016; Von Oswald et al., 2020; D'Angelo & Fortuin, 2021; Wenzel et al., 2020; D'Angelo & Fortuin, 2021).

Inspired by these developments, we make use of the technique proposed in D'Angelo & Fortuin (2021) where the members of an ensemble interact with each other through a repulsive force that encourages diversity in the ensemble. Particularly, we explore three choices for repulsion loss:

$$\mathcal{L}_{repulsion} = \begin{cases} \mathcal{L}_{repulsion}^{\text{IL2}} = \frac{1}{||\theta - \theta_N||_2^2} & \text{(Inverse } \ell2 \text{ loss)} \\ \mathcal{L}_{repulsion}^{\text{NL2}} = -||\theta - \theta_N||_2^2 & \text{(Negative } \ell2 \text{ loss)} \\ \mathcal{L}_{repulsion}^{\text{EL2}} = \exp(-\alpha||\theta - \theta_N||_2^2) & \text{(Exponential negative } \ell2 \text{ loss)} \end{cases} \qquad (6)$$

It can be seen that minimization of all of these choices will force $\theta$ to be away from $\theta_N$, consequently serving our purpose. In fact, in general, one can use any function of $||\theta - \theta_N||_2^2$ that has a global maxima at $\theta_N$ as a choice for repulsion loss. In this work, we work with the above mentioned choices. An algorithmic overview of Stage-1 Negative Adaptation is presented in Algorithms 11 and Stage-2 Unlearning is presented in Algorithm 6

## 4 Theoretical Discussion

In this section, we present theoretical insights into the proposed method. Inspired by the work in Nguyen et al. (2020), we operate in Bayesian setting for these claims and make use of widely used Laplace approximation around relevant parameters. Specifically, we demonstrate that for an optimal discriminator, the proposed regularization term combined with the adversarial term results in a contrastive divergence-like objective (a difference of two divergence terms). This encourages the generator to capture the implicit distribution of the pre-trained generator without the support of negative samples while maximizing the divergence between the parameter distribution of the post-unlearning generator and that of the generator which produces negative samples (Theorem 1). This result is shown in . Further, we show the relation between the parameter space divergence and data space divergence (Claim 1).

For this, let $\Theta$ denote the parameter space of a generator network. Let $\theta_G \in \Theta$ be the parameter of a pre-trained generator with an implicit distribution $p_G^{\mathcal{X}}(x)$ over the data space[1]. Further, consider two distributions $p_N^{\Theta}(\theta)$ and $p_U^{\Theta}(\theta)$ over $\Theta$, where the latter is a learnable distribution and the former is such that for $\mathbf{z} \sim p_Z(z)$ the corresponding generated samples from manifested generator $G_\theta(\mathbf{z}) \sim p_N^{\mathcal{X}}(x)$. In other words, samples from $p_N^{\Theta}(\theta)$ lead to the generation of negative samples.

**Theorem 1.** *Consider the distributions $p_N^{\Theta}(\theta)$ and $p_U^{\Theta}(\theta)$ to be Gaussian, i.e., $p_N^{\Theta}(\theta) = \frac{1}{|2\pi\Sigma|^{d/2}} \exp\left[\frac{1}{2}(\theta - \theta_N)^T \Sigma^{-1}(\theta - \theta_N)\right]$ and $p_U^{\Theta}(\theta) = \frac{1}{|2\pi\Sigma|^{d/2}} \exp\left[\frac{1}{2}(\theta - \theta_P)^T \Sigma^{-1}(\theta - \theta_P)\right]$, where $\Sigma = I$, $\theta_N$ and $\theta_P$ are the mean parameters and $\theta_P$ is learnable. Then statements (1 - 3) hold for the following optimization problem:*

$$\min_{\theta_P} \max_{\phi} \mathbb{E}_{\mathbf{x} \sim p_{G \setminus N}(x)} [\log D_\phi(\mathbf{x})] + \mathbb{E}_{\substack{\mathbf{z} \sim p_Z \\ \theta \sim p_U}} [\log(1 - D_\phi(G_\theta(\mathbf{z})))] + \mathcal{L}_{repulsion} \qquad (7)$$

1. *for $\mathcal{L}_{repulsion} = \mathcal{L}_{repulsion}^{IL2}$, solving Eq. 7 leads to $p_U^{\Theta}$ that minimizes*

$$D_{JSD}(p_{G \setminus N}^{\mathcal{X}} \,||\, p_U^{\mathcal{X}}) + \left[D_{KL}(p_U^{\Theta} \,||\, p_N^{\Theta})\right]^{-1}$$

2. *for $\mathcal{L}_{repulsion} = \mathcal{L}_{repulsion}^{NL2}$, solving Eq. 7 leads to $p_U^{\Theta}$ that minimizes*

$$D_{JSD}(p_{G \setminus N}^{\mathcal{X}} \,||\, p_U^{\mathcal{X}}) - D_{KL}(p_U^{\Theta} \,||\, p_N^{\Theta})$$

3. *for $\mathcal{L}_{repulsion} = \mathcal{L}_{repulsion}^{EL2}$, solving Eq. 7 leads to $p_U^{\Theta}$ that minimizes*

$$D_{JSD}(p_{G \setminus N}^{\mathcal{X}} \,||\, p_U^{\mathcal{X}}) - D_H(p_U^{\Theta} \,||\, p_N^{\Theta})$$

---

[1]For convenience, we use superscript $\mathcal{X}$ and $\Theta$ to denote a distribution in data space and parameter space respectively. We use the data space distribution from previous section as it is with a superscript $\mathcal{X}$ for this distinction.

*where, $D_{KL}(\cdot \mid\mid \cdot)$ and $D_H(\cdot \mid\mid \cdot)$ denote KL divergence and Hellinger divergence, and $G_\theta(\mathbf{z}) \sim p_U^{\mathcal{X}}(\cdot)$ denotes the implicit distribution of generator when $\mathbf{z} \sim p_Z$ and $\theta \sim p_U^\Theta$.*

The proof for the aforementioned result is relatively straightforward; for the sake of completeness, we include the proof in the Appendix. It is evident from the above result that the unlearning phase of the proposed method, assuming a Gaussian parameter distribution, achieves two objectives: (a) minimizing the Jensen-Shannon Divergence between the learned data distribution ($p_U^{\mathcal{X}}$) and the implicit generator distribution after removing the support of negative samples ($p_{G \setminus N}^{\mathcal{X}}$), and (b) maximizing a suitable divergence measure between the parameter distribution during unlearning ($p_U^\Theta$) and the parameter distribution that leads to the generation of negative samples ($p_N^\Theta$). While (a) is relatively straightforward, (b) provides valuable insights into the effect of the regularization term, which is particularly interesting. The regularization term ensures that the current parameter distribution moves away from the one responsible for generating undesired features, effectively aligning with the primary goal of unlearning. Furthermore, this behavior has been shown to be crucial for unlearning in Bayesian settings, as demonstrated in Nguyen et al. (2020). Essentially, this aligns with the desired outcome of unlearning, ensuring that the model captures only the desired support of the distribution.

An interesting observation from the above result is that utilizing $\mathcal{L}_{\text{repulsion}}^{\text{NL2}}$ or $\mathcal{L}_{\text{repulsion}}^{\text{EL2}}$ results in an objective akin to contrastive divergence, i.e., it entails the difference between two divergences. However, these two divergence metrics operate on distributions in distinct spaces: the first divergence operates in the data space, while the second operates in parameter space. This prompts a natural question: how does the divergence in parameter space relate to the corresponding distribution in data space? We address this question in the following claim.

**Claim 1.** *For any general $f$-divergence $D_f(\cdot \mid \cdot)$, and a given latent vector, the following inequality holds:*

$$D_{JSD}(p_{G \setminus N}^{\mathcal{X}} \mid\mid p_U^{\mathcal{X}}) - D_f(p_U^\Theta \mid\mid p_N^\Theta) \leq D_{JSD}(p_{G \setminus N}^{\mathcal{X}} \mid\mid p_U^{\mathcal{X}}) - D_f(p_U^{\mathcal{X}} \mid\mid p_N^{\mathcal{X}}) \tag{8}$$

Above result relies on simple application of data-processing inequality. The proof is provided in Appendix. Since, KL and Hellinger divergence are both instances of $f$-divergence, the above result holds for Statements 2 and 3 of Theorem 1. Hence, we see that the while using $\mathcal{L}_{repulsion}^{\text{NL2}}$ or $\mathcal{L}_{repulsion}^{\text{EL2}}$, the corresponding data space objectives act as upper bounds to the parameter space objectives. With these insights, we end the theoretical discussion.

## 5 Experiments and Results

### 5.1 Datasets

An unlearning algorithm should ensure that the generator should not generate images containing the undesired (or unlearnt) feature. For our experiments, we consider two types of unlearning settings: (i) Class-level unlearning and (ii) Feature-level unlearning. The primary difference between the two type of unlearning lies in the nature of the associations. In feature-level unlearning, an image can exhibit multiple features simultaneously, whereas in class-level unlearning, an image from one class cannot belong to any other class. In other words, if each feature is treated as a class, feature-level unlearning allows an image to belong to multiple classes, while class-level unlearning restricts an image to a single class. For instance, a person with bangs can be either male or female, but a digit labeled as 'one' cannot simultaneously belong to any other class.

We use MNIST dataset (LeCun et al., 1998) and AFHQ dataset (Choi et al., 2020) for class-level unlearning. MNIST consists of $60,000$ $28 \times 28$ dimensional black and white images of handwritten digits. For our experiments, we take three-digit classes: 1, 4, and 8 for unlearning. AFHQ consists of $15,000$ high-quality animal face images at $512 \times 512$ resolution with three categories: cat, dog and wildlife. We unlearn each class one at a time in our experiments. Similarly, we use CelebA-HQ dataset (Liu et al., 2015) for feature-level unlearning. CelebA-HQ contains $30,000$ RGB high-quality celebrity face images of dimension $256 \times 256$. Here, we unlearn the following subtle features: (a) Bangs, (b) Hats, (c) Bald, and (d) Eyeglasses.

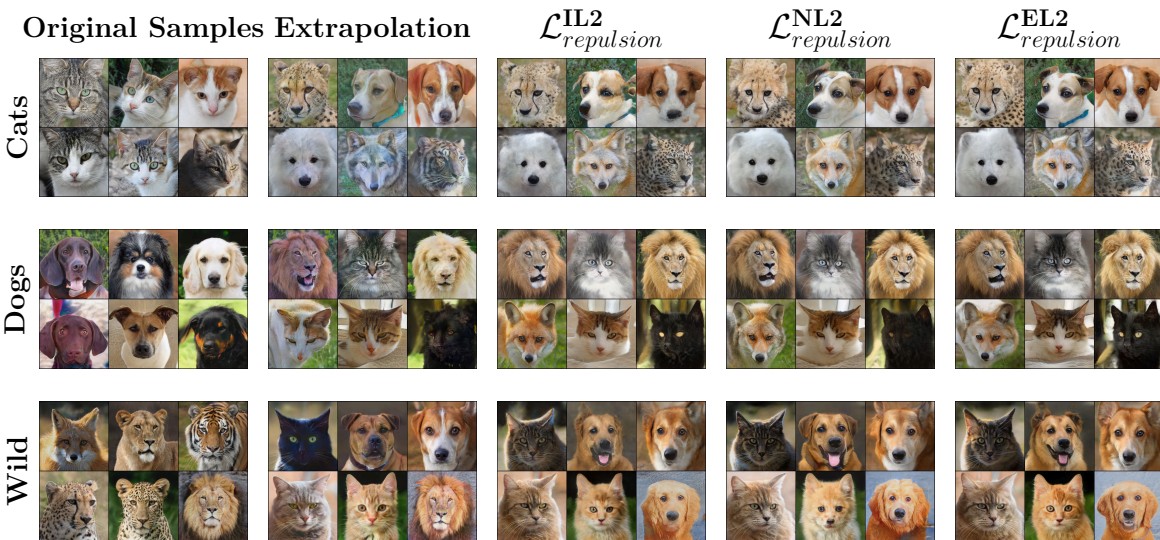

Figure 2: Results of Unlearning different classes on AFHQ dataset.

## 5.2 Experimental Details

**Training Details:** We use one of the state-of-the-art and widely used high-fidelity StyleGAN2 (Karras et al., 2020) for demonstrating the performance of the proposed method on the tasks mentioned in previous section. The StyleGAN is trained on the entire MNIST, AFHQ and CelebA-HQ datasets to obtain the pre-trained GAN from which we desire to unlearn specific features. The training details of StyleGAN2 are given in Supplementary Section A.

**Unlearning Details**: In our experiments, we employ a pre-trained classifier as a proxy for human to obtain user feedback. Specifically, we pre-train the classifier to classify a given image as desired or undesired (depending upon the feature under consideration). We classify $1,000$ generated images from pre-trained GAN as positive and negative samples using the pre-trained classifier. The generated samples containing the undesired features are marked as negative samples and the rest of the images are marked as positive samples. These samples are then used in Stage-1 and Stage-2 of the proposed method for unlearning as described in Section 3. We evaluate our result using all the choices of repulsion loss as mentioned in Eq. 6. For reproducibility, we provide all the hyper-parameters and training details in Supplementary Section A. The original FID of the GAN after training is as follows- MNIST: 5.4, AFHQ: 8.1, CelebA-HQ: 5.3. We mention these in caption of each table wherever necessary.

## 5.3 Baselines and Evaluation Metrics

**Baselines**: To the best of our knowledge, ours is one of the first works that addresses the problem of unlearning in *high-fidelity* generator models such as StyleGAN2. Hence, we evaluate and compare our method with all the candidates for repulsion loss presented in Eq. 6. Further, we also include the results with extrapolation in the parameter space as demonstrated in figure 1 (b). We include recent unlearning baselines tailored for classification, easily adaptable to generative tasks. Specifically, we incorporate EU-$k$, CF-$k$, and $\ell 1$-sparse (Goel et al., 2022; Jia et al., 2023) for comparison, with detailed information in the Supplementary section A.4. Additionally, we assess our method against GAN adaptation to positive samples, utilizing recent generative few-shot adaptation methods like EWC, CDC, and RSSA (Li et al., 2020; Ojha et al., 2021; Xiao et al., 2022) as baselines. Apart from the above baselines, we also mention the results obtained from training a GAN from scratch on only desirable data present in the dataset. This model acts as the gold standard, however, due to unavailability of underlying dataset, this is not practical. Nonetheless, we mention it in our tables for completeness. We evaluate the performance of each method across three independent runs and report the result in the form of `mean ± std. dev`.

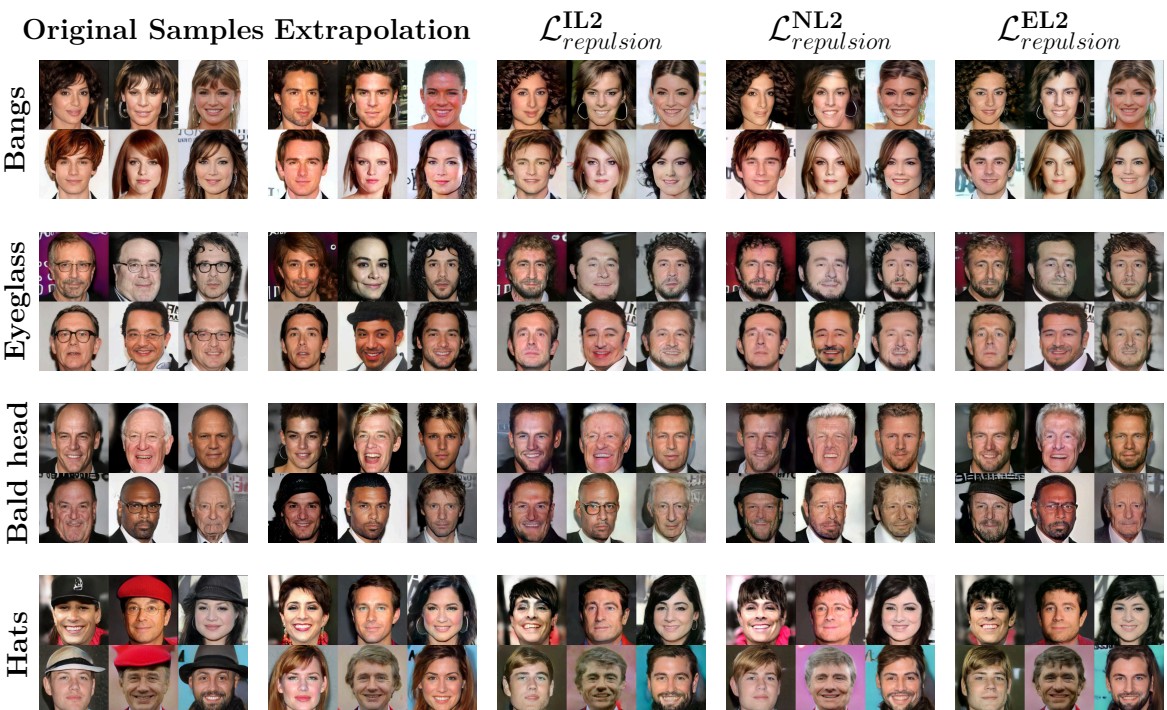

Figure 3: Results of Unlearning different features on CelebA dataset.

**Evaluation Metrics**: Various metrics have been devised for assessing machine unlearning methods (Xu et al., 2020). To gauge the effectiveness of our proposed techniques and the baseline methods, we utilize three fundamental evaluation metrics:

1. **Percentage of Un-Learning (PUL)**: This metric quantifies the extent of unlearning by measuring the reduction in the number of negative samples generated by the GAN post-unlearning compared to the pre-unlearning state. PUL is computed as: $\text{PUL} = \frac{(S_n)_{\theta_G} - (S_n)_{\theta_P}}{(S_n)_{\theta_G}} \times 100$, where, $(S_n)_{\theta_G}$ and $(S_n)_{\theta_P}$ represent the number of negative samples generated by the original GAN and the GAN after unlearning respectively. We generate 15,000 random samples from both GANs and employ a pre-trained classifier (as detailed in Section 5.2) to identify the negative samples. PUL provides a quantitative measure of the extent of the unlearning algorithm in eliminating the undesired feature from the GAN.

2. **Fréchet Inception Distance (FID)**: While PUL quantifies the degree of unlearning, it does not assess the quality of samples generated by the GAN post-unlearning. Hence, we calculate the FID (Heusel et al., 2017) between the generated samples and the original dataset without the undesired samples.

3. **Retraining FID (Ret-FID)**: To resemble the retrained GAN, we compute the FID between the outputs of the GAN after unlearning and the GAN trained from scratch on the dataset obtained after eliminating undesired features.

Please note that the original dataset is unavailable during the unlearning process. Consequently, the use of the original dataset is solely for evaluation purposes.

## 5.4 Unlearning Results

We present our results and observations on MNIST, AFHQ and CelebA-HQ in Table 1, 2, 3 respectively. We observe that the choice of $\mathcal{L}_{repulsion}^{\text{EL2}}$ as repulsion loss provides the highest PUL in most of the cases

Table 1: PUL ($\uparrow$), FID ($\downarrow$), and Ret-FID ($\downarrow$) after unlearning MNIST classes. FID of pre-trained GAN: 5.4.

| Method | Class 1 | | | Class 4 | | | Class 8 | | |
|---|---|---|---|---|---|---|---|---|---|
| | PUL | FID | Ret-FID | PUL | FID | Ret-FID | PUL | FID | Ret-FID |
| Retraining | $98.80 \pm 0.09$ | $4.94 \pm 0.04$ | N/A | $98.58 \pm 0.28$ | $5.24 \pm 0.09$ | N/A | $99.72 \pm 0.01$ | $4.80 \pm 0.11$ | N/A |
| CF-$k$ | $18.60 \pm 2.30$ | $92.88 \pm 0.51$ | $89.05 \pm 0.93$ | $7.71 \pm 0.18$ | $33.61 \pm 0.93$ | $30.76 \pm 0.86$ | $16.03 \pm 0.09$ | $36.67 \pm 1.98$ | $33.14 \pm 0.74$ |
| EU-$k$ | $31.77 \pm 1.56$ | $15.34 \pm 0.01$ | $14.05 \pm 0.17$ | $17.59 \pm 3.31$ | $8.80 \pm 0.08$ | $7.84 \pm 0.24$ | $21.66 \pm 0.70$ | $8.92 \pm 0.77$ | $7.10 \pm 0.10$ |
| $\ell$1-Sparse | $91.84 \pm 0.55$ | $22.17 \pm 0.14$ | $17.87 \pm 0.42$ | $94.16 \pm 0.06$ | $23.24 \pm 0.59$ | $17.25 \pm 0.22$ | $95.42 \pm 0.01$ | $22.23 \pm 0.05$ | $16.83 \pm 0.13$ |
| EWC | $90.93 \pm 0.46$ | $7.18 \pm 0.08$ | $5.10 \pm 0.01$ | $82.78 \pm 1.02$ | $9.34 \pm 0.08$ | $5.91 \pm 0.04$ | $92.70 \pm 0.81$ | $9.35 \pm 0.04$ | $6.03 \pm 0.04$ |
| CDC | $90.70 \pm 0.16$ | $20.85 \pm 0.49$ | $17.51 \pm 1.86$ | $42.39 \pm 0.66$ | $17.82 \pm 1.99$ | $12.39 \pm 2.32$ | $18.86 \pm 0.92$ | $11.18 \pm 2.28$ | $12.38 \pm 3.73$ |
| RSSA | $38.36 \pm 0.09$ | $12.70 \pm 0.17$ | $13.54 \pm 0.40$ | $71.96 \pm 0.21$ | $25.41 \pm 0.06$ | $19.08 \pm 0.59$ | $79.54 \pm 0.32$ | $34.59 \pm 0.26$ | $25.33 \pm 0.39$ |
| Extrapolation | $95.10 \pm 0.69$ | $41.39 \pm 1.76$ | $42.98 \pm 0.68$ | $94.50 \pm 0.05$ | $17.90 \pm 0.35$ | $27.81 \pm 0.37$ | $90.90 \pm 0.12$ | $45.79 \pm 0.29$ | $44.30 \pm 0.40$ |
| $\mathcal{L}_{\text{repulsion}}^{\text{NL2}}$ (Ours) | $97.85 \pm 2.25$ | $9.69 \pm 0.07$ | $6.70 \pm 0.25$ | $93.03 \pm 0.70$ | $10.50 \pm 0.34$ | $6.26 \pm 0.12$ | $97.92 \pm 0.67$ | $9.95 \pm 0.17$ | $6.70 \pm 0.18$ |
| $\mathcal{L}_{\text{repulsion}}^{\text{IL2}}$ (Ours) | $92.97 \pm 0.48$ | $13.06 \pm 0.46$ | $16.55 \pm 0.54$ | $90.39 \pm 1.36$ | $15.54 \pm 0.05$ | $8.64 \pm 0.90$ | $\mathbf{98.28 \pm 0.55}$ | $9.72 \pm 0.31$ | $11.64 \pm 0.46$ |
| $\mathcal{L}_{\text{repulsion}}^{\text{EL2}}$ (Ours) | $\mathbf{99.32 \pm 0.43}$ | $9.65 \pm 0.21$ | $6.29 \pm 0.18$ | $\mathbf{96.23 \pm 0.54}$ | $10.24 \pm 0.19$ | $5.80 \pm 0.04$ | $95.22 \pm 0.55$ | $8.89 \pm 0.52$ | $5.68 \pm 0.10$ |

Table 2: PUL ($\uparrow$), FID ($\downarrow$), and Ret-FID ($\downarrow$) after unlearning AFHQ classes. FID of pre-trained GAN: 8.1.

| Method | Cat | | | Dog | | | Wild | | |
|---|---|---|---|---|---|---|---|---|---|
| | PUL | FID | Ret-FID | PUL | FID | Ret-FID | PUL | FID | Ret-FID |
| Retraining | $93.37 \pm 0.23$ | $12.45 \pm 0.16$ | N/A | $85.96 \pm 0.21$ | $5.71 \pm 0.43$ | N/A | $88.60 \pm 0.22$ | $15.24 \pm 0.24$ | N/A |
| CF-$k$ | $15.74 \pm 0.41$ | $42.91 \pm 0.33$ | $37.34 \pm 0.29$ | $13.12 \pm 0.12$ | $64.50 \pm 0.58$ | $65.72 \pm 2.31$ | $16.40 \pm 0.55$ | $35.56 \pm 0.11$ | $34.47 \pm 0.94$ |
| EU-$k$ | $16.08 \pm 0.28$ | $43.21 \pm 0.25$ | $37.62 \pm 0.42$ | $10.30 \pm 0.13$ | $32.69 \pm 0.23$ | $31.40 \pm 0.36$ | $16.63 \pm 0.84$ | $35.80 \pm 0.39$ | $32.55 \pm 0.14$ |
| $\ell$1-Sparse | $86.25 \pm 1.35$ | $25.73 \pm 0.05$ | $14.96 \pm 0.11$ | $68.21 \pm 0.15$ | $19.54 \pm 0.53$ | $15.10 \pm 0.85$ | $84.05 \pm 0.45$ | $37.63 \pm 0.52$ | $21.80 \pm 0.16$ |
| EWC | $90.58 \pm 1.54$ | $15.30 \pm 0.98$ | $8.29 \pm 0.04$ | $68.79 \pm 0.08$ | $10.38 \pm 1.08$ | $8.50 \pm 0.35$ | $73.44 \pm 0.16$ | $16.32 \pm 0.08$ | $14.38 \pm 0.01$ |
| CDC | $28.57 \pm 0.43$ | $59.76 \pm 0.40$ | $49.93 \pm 1.02$ | $9.60 \pm 0.24$ | $47.85 \pm 0.53$ | $46.79 \pm 0.67$ | $6.51 \pm 0.13$ | $42.17 \pm 0.19$ | $36.95 \pm 0.28$ |
| RSSA | $66.86 \pm 0.22$ | $59.88 \pm 0.35$ | $46.80 \pm 0.62$ | $54.54 \pm 0.36$ | $55.06 \pm 0.20$ | $53.24 \pm 0.34$ | $43.99 \pm 0.72$ | $60.95 \pm 0.45$ | $44.20 \pm 0.32$ |
| Extrapolation | $90.64 \pm 0.33$ | $45.89 \pm 2.60$ | $38.87 \pm 1.93$ | $82.08 \pm 1.02$ | $25.54 \pm 1.19$ | $24.98 \pm 1.35$ | $84.45 \pm 0.11$ | $45.98 \pm 2.66$ | $41.86 \pm 1.99$ |
| $\mathcal{L}_{\text{repulsion}}^{\text{NL2}}$ (Ours) | $94.28 \pm 0.17$ | $16.29 \pm 0.06$ | $8.93 \pm 1.23$ | $75.33 \pm 0.35$ | $8.62 \pm 0.08$ | $5.96 \pm 0.45$ | $82.82 \pm 0.77$ | $17.69 \pm 0.18$ | $14.89 \pm 0.13$ |
| $\mathcal{L}_{\text{repulsion}}^{\text{IL2}}$ (Ours) | $90.93 \pm 0.51$ | $20.69 \pm 0.04$ | $9.86 \pm 0.08$ | $76.53 \pm 0.51$ | $9.37 \pm 1.06$ | $6.84 \pm 0.10$ | $80.96 \pm 0.18$ | $22.70 \pm 0.11$ | $13.76 \pm 0.12$ |
| $\mathcal{L}_{\text{repulsion}}^{\text{EL2}}$ (Ours) | $\mathbf{95.76 \pm 0.25}$ | $16.50 \pm 0.12$ | $8.17 \pm 0.14$ | $\mathbf{79.21 \pm 0.18}$ | $9.31 \pm 0.14$ | $7.13 \pm 0.12$ | $\mathbf{89.09 \pm 0.25}$ | $19.67 \pm 0.69$ | $14.90 \pm 0.65$ |

Table 3: PUL ($\uparrow$), FID ($\downarrow$), and Ret-FID ($\downarrow$) after unlearning CelebA-HQ features. FID of pre-trained GAN: 5.3.

| Method | Bangs | | | Hat | | | Bald | | | Eyeglasses | | |
|---|---|---|---|---|---|---|---|---|---|---|---|---|
| | PUL | FID | Ret-FID | PUL | FID | Ret-FID | PUL | FID | Ret-FID | PUL | FID | Ret-FID |
| Retraining | $84.47 \pm 1.49$ | $7.58 \pm 0.06$ | N/A | $98.65 \pm 0.03$ | $6.35 \pm 0.10$ | N/A | $72.13 \pm 0.07$ | $7.18 \pm 0.08$ | N/A | $58.57 \pm 0.04$ | $6.50 \pm 0.77$ | N/A |
| CF-$k$ | $18.60 \pm 2.30$ | $9.65 \pm 0.15$ | $7.70 \pm 0.27$ | $15.22 \pm 0.05$ | $9.58 \pm 0.09$ | $7.57 \pm 0.05$ | $48.17 \pm 1.76$ | $9.30 \pm 0.12$ | $7.37 \pm 0.06$ | $16.41 \pm 0.73$ | $9.49 \pm 0.15$ | $6.79 \pm 0.06$ |
| EU-$k$ | $20.08 \pm 1.35$ | $9.03 \pm 0.23$ | $7.38 \pm 0.06$ | $17.96 \pm 1.54$ | $9.40 \pm 0.21$ | $7.16 \pm 0.05$ | $52.18 \pm 0.53$ | $9.03 \pm 0.06$ | $6.92 \pm 0.05$ | $16.41 \pm 1.33$ | $9.39 \pm 0.04$ | $6.85 \pm 0.85$ |
| $\ell$1-Sparse | $75.78 \pm 1.48$ | $16.20 \pm 0.24$ | $13.77 \pm 0.46$ | $59.01 \pm 1.67$ | $8.61 \pm 0.09$ | $5.66 \pm 0.09$ | $67.26 \pm 0.19$ | $16.27 \pm 0.73$ | $14.14 \pm 0.59$ | $82.30 \pm 1.96$ | $15.95 \pm 0.02$ | $12.36 \pm 0.32$ |
| EWC | $76.24 \pm 2.14$ | $9.64 \pm 0.05$ | $6.69 \pm 0.03$ | $74.14 \pm 0.78$ | $9.06 \pm 0.22$ | $6.44 \pm 0.08$ | $55.70 \pm 2.57$ | $9.32 \pm 0.07$ | $6.77 \pm 0.04$ | $46.51 \pm 0.57$ | $9.19 \pm 0.04$ | $6.56 \pm 0.01$ |
| CDC | $83.76 \pm 0.12$ | $34.66 \pm 1.94$ | $24.67 \pm 0.02$ | $85.23 \pm 0.77$ | $23.48 \pm 0.12$ | $17.46 \pm 0.22$ | $90.10 \pm 0.37$ | $26.96 \pm 0.02$ | $17.99 \pm 0.09$ | $91.65 \pm 0.31$ | $36.66 \pm 0.02$ | $30.08 \pm 0.27$ |
| RSSA | $32.07 \pm 0.58$ | $17.79 \pm 0.02$ | $15.99 \pm 0.08$ | $22.70 \pm 1.47$ | $18.83 \pm 0.08$ | $14.87 \pm 0.16$ | $31.70 \pm 1.89$ | $20.35 \pm 0.34$ | $18.63 \pm 0.01$ | $26.21 \pm 0.17$ | $22.24 \pm 0.16$ | $18.09 \pm 0.02$ |
| Extrapolation | $89.54 \pm 0.09$ | $11.54 \pm 0.07$ | $11.02 \pm 0.06$ | $94.35 \pm 0.12$ | $12.18 \pm 0.04$ | $10.12 \pm 0.07$ | $94.44 \pm 0.34$ | $23.44 \pm 0.02$ | $26.40 \pm 0.30$ | $92.80 \pm 0.14$ | $23.70 \pm 0.07$ | $19.10 \pm 0.10$ |
| $\mathcal{L}_{\text{repulsion}}^{\text{NL2}}$ (Ours) | $90.41 \pm 0.19$ | $11.92 \pm 0.46$ | $8.69 \pm 0.05$ | $93.99 \pm 1.70$ | $9.60 \pm 0.25$ | $6.44 \pm 0.11$ | $\mathbf{97.13 \pm 1.42}$ | $14.70 \pm 0.55$ | $9.03 \pm 0.13$ | $83.76 \pm 3.21$ | $12.81 \pm 0.88$ | $7.93 \pm 0.99$ |
| $\mathcal{L}_{\text{repulsion}}^{\text{IL2}}$ (Ours) | $84.05 \pm 1.03$ | $13.09 \pm 0.10$ | $9.07 \pm 0.18$ | $94.00 \pm 0.75$ | $11.31 \pm 0.06$ | $7.25 \pm 0.13$ | $83.51 \pm 2.18$ | $12.94 \pm 0.89$ | $9.87 \pm 0.04$ | $75.23 \pm 6.25$ | $13.12 \pm 0.78$ | $6.11 \pm 0.24$ |
| $\mathcal{L}_{\text{repulsion}}^{\text{EL2}}$ (Ours) | $\mathbf{90.45 \pm 1.02}$ | $11.16 \pm 0.08$ | $7.94 \pm 0.32$ | $\mathbf{94.40 \pm 2.19}$ | $9.45 \pm 0.96$ | $6.31 \pm 0.64$ | $93.97 \pm 2.65$ | $11.07 \pm 0.86$ | $7.83 \pm 0.05$ | $\mathbf{93.63 \pm 0.42}$ | $9.66 \pm 0.58$ | $9.84 \pm 0.23$ |

for both the datasets. Further, it also provides the best FID and Ret-FID as compared to other choices of repulsion loss. $\mathcal{L}_{repulsion}^{\text{NL2}}$ is stands out to be the second best in these metrics for most of the cases. Further, we observe that across all datasets, the classification unlearning baselines perform very poorly on all metrics for unlearning in GANs. This tells us that methods proposed for unlearning in classification are not suited for unlearning in generative tasks. And lastly, we find that few-shot adaptation baselines, give relatively poor results when compared to the proposed method. This observation indicates that it is not enough to just adapt the GAN on the positive samples for unlearning, one needs to go further and use additional regularization to unlearn the undesired features.

For MNIST, we observe in Table 1 that the proposed method with $\mathcal{L}_{repulsion}^{\text{EL2}}$ as repulsion loss consistently provides a PUL of above 95% while giving the best FID and Ret-FID compared to other methods. We also observe that Extrapolation in parameter space leads to significant PUL albeit the FID and Ret-FID

are considerably worse compared to proposed method under different repulsion loss. This shows that the proposed method decently solves the task of unlearning at class-level.

We make similar observations for high-resolution AFHQ dataset as well. One can see that the proposed method provides highest PUL in all the cases, while maintaining the FID as well as Ret-FID. We observe highest PUL while unlearning the 'Cat' class while lowest PUL is observed in 'Dog' class.

Lastly, for feature-level unlearning results on CelebA-HQ, it can be seen that the proposed method with $\mathcal{L}_{repulsion}^{\text{EL2}}$ as repulsion loss consistently provides a PUL of above 90%, illustrating significant unlearning of undesired features. Further, the FID and Ret-FID using $\mathcal{L}_{repulsion}^{\text{EL2}}$ stand out to be the best among all the methods with significant PUL.

We also observe some drop in FID across all dataset after unlearning. E.g., the FID of the samples generated by the unlearnt GAN (on Hats) using $\mathcal{L}_{repulsion}^{\text{EL2}}$ drops by about 4.15 points while it drops by 4.3 and 6.01 points while using $\mathcal{L}_{repulsion}^{\text{NL2}}$ and $\mathcal{L}_{repulsion}^{\text{IL2}}$ as compared to the pre-trained GAN. On the other hand, Extrapolation in parameter space leads to a drop of 6.88 points in FID. This further validates the need of repulsion regularizer to maintain the generation quality. This observation is consistent across all datasets and features. This supports our claim that extrapolation might unlearn the undesired feature, however, it deteriorates the quality of generated samples significantly.

Another interesting observation from the above results is that the classification unlearning baselines consistently provide lower PUL, albeit with slightly better FID and Ret-FID. This tells us that these baselines while capable of unlearning in classification tasks, fail to nudge the generator appropriately for desired unlearning task. Leading to a suboptimal generator which still generates undesired samples, without compromising on the quality of the generated samples.

The visual illustration of these methods for AFHQ and CelebA-HQ are shown in figure 2 and figure 3 respectively. Here, we observe that the proposed method effectively unlearns the undesired feature. Moreover, it can be seen that the unlearning through extrapolation leads to the unlearning of correlated features as well. E.g. Bangs are correlated with female attributes. It can be seen that the unlearning of Bangs through extrapolation also leads to the unlearning of female feature which is not desired. However, unlearning through the proposed method unlearns Bangs only, while keeping the other features as it is. Similar observations could be made for AFHQ, where extrapolation leads to some minor artifacts in generates samples, whereas proposed method generates plausible images without any artifacts. Similar visual results for MNIST is provided in Supplementary Section B. We also provide visualization of random samples generated from original GAN and the GAN after unlearning in Supplementary Section F to give a qualitative idea of the generation quality.

Another aspect of unlearning that we explore in our work is the effect of unlearning on other features. Particularly, unlearning an undesired feature should not disturb the other features. For this we generate samples from pre-trained and post-unlearning GANs. Then, we calculate the occurrence of specific features within the two GANs and report the percentage change in these numbers. These results could be found in Section E of Supplementary. As discussed in Section 3, the approach of first adapting the model to negative samples followed by applying the repulsion loss during the Unlearning phase can also be extended to scenarios where curated datasets of positive and negative samples are available. We demonstrate this in Section D.

## 5.5 Comparison with DC-GAN Baselines

As previously mentioned, our proposed method operates with high-fidelity GANs. Nonetheless, in addition to the tailored baselines, a few previous methods aim to unlearn undesired features in more primitive GANs, such as DC-GANs, on low-resolution images. Notably, the methods proposed in Sun et al. (2023) and Kong & Chaudhuri (2023) are highly relevant to our work. However, they primarily operate on DC-GAN. To ensure a fair comparison, we implement our method on DC-GAN and evaluate it against these baselines using the MNIST dataset (with $\mathcal{L}_{repulsion}^{\text{EL2}}$). Specifically, we compare our approach to the cascaded unlearning algorithm (CUA) from Sun et al. (2023) and the data redaction method using validity data (DRed) from Kong & Chaudhuri (2023). Our findings are summarized in Table 4. The results indicate that our method

Table 4: Comparison of the proposed method against DC-GAN baselines

| Method | Metrics | Class 1 | Class 4 | Class 8 |
|--------|---------|---------|---------|---------|
| CUA | PUL | $96.12 \pm 1.21$ | $95.49 \pm 0.37$ | $97.34 \pm 1.21$ |
| | FID | $12.73 \pm 0.48$ | $11.61 \pm 1.81$ | $13.09 \pm 1.03$ |
| | Ret-FID | $11.02 \pm 1.52$ | $11.57 \pm 0.79$ | $10.09 \pm 1.21$ |
| DRed | PUL | $98.13 \pm 0.12$ | $97.70 \pm 0.26$ | $96.54 \pm 0.23$ |
| | FID | $11.72 \pm 1.29$ | $12.82 \pm 1.11$ | $10.32 \pm 0.82$ |
| | Ret-FID | $8.72 \pm 0.72$ | $8.93 \pm 0.86$ | $10.12 \pm 1.41$ |
| Ours | PUL | $\mathbf{98.76 \pm 0.56}$ | $\mathbf{99.18 \pm 0.28}$ | $\mathbf{98.72 \pm 0.42}$ |
| | FID | $\mathbf{10.37 \pm 0.94}$ | $\mathbf{9.72 \pm 1.33}$ | $\mathbf{10.27 \pm 0.82}$ |
| | Ret-FID | $\mathbf{6.32 \pm 0.43}$ | $\mathbf{7.23 \pm 1.28}$ | $\mathbf{7.66 \pm 1.30}$ |

Table 5: Effect on PUL (↑), FID (↓), and Ret-FID (↓) with and without repulsion loss.

| Features | Metrics | $\mathcal{L}'_{adv}$ | $\mathcal{L}'_{adv} + \mathcal{L}^{\mathbf{EL2}}_{repulsion}$ |
|----------|---------|------|------|
| Bangs | PUL | $79.89 \pm 0.49$ | $\mathbf{90.45 \pm 1.01}$ |
| | FID | $\mathbf{10.06 \pm 0.24}$ | $11.16 \pm 0.08$ |
| | Ret-FID | $8.69 \pm 0.04$ | $\mathbf{7.94 \pm 0.32}$ |
| Hat | PUL | $84.68 \pm 3.89$ | $\mathbf{94.40 \pm 2.19}$ |
| | FID | $9.66 \pm 0.16$ | $\mathbf{9.45 \pm 0.96}$ |
| | Ret-FID | $6.45 \pm 0.08$ | $\mathbf{6.04 \pm 0.02}$ |

outperforms both baselines across all metrics in all scenarios. All methods performed well on PUL, with our method achieving the best PUL, followed by DRed and then CUA in most cases. A similar trend is observed in the FID scores. Although DRed is a close competitor to our proposed method, our approach yields a significantly better Ret-FID than DRed, suggesting that the post-unlearning GAN using our method is closer to the gold standard compared to DRed.

## 5.6 Ablation Study

Lastly, we present the ablation study to observe the effect of repulsion loss. In particular, we see if adapting the pre-trained GAN only on the positive samples leads to desired levels of unlearning. Our observations on CelebA-HQ for Bangs and Hats are presented in Table 5. Here, we use $\mathcal{L}^{\mathrm{EL2}}_{repulsion}$ as repulsion loss. It can be seen that only using adversarial loss doesn't lead to significant unlearning of undesired feature. E.g. using repulsion loss provides and increase of about 10.56% and 9.72% in PUL. The FID increases by minor 0.66 point on Bangs while it decreases by 0.21 points on Hats. Hence, we conclude that repulsion loss is indeed crucial for unlearning.

## 6 Conclusion

We propose a novel unlearning method designed for high-fidelity GANs. Our approach is distinguished via its unique ability to operate in zero-shot scenario, entirely independent of the original data on which GAN is trained. We operate under feedback-based framework in two stages. The initial stage adapts the pre-trained GAN on the negative samples whereas the later stage unlearns the undesired feature by adapting on positive samples along with a repulsion regularizer. A notable advantage of our approach is its capability to conduct the unlearning process without significantly impacting other desirable features. We firmly believe that our work represents a substantial advancement in the field of unlearning within deep generative models. This progress holds particular relevance in addressing critical societal concerns, particularly those related to the generation of biased, racial, or harmful content by these models.

**Limitation and Future Work**: We note that our study does not address aspects like 'toxicity' due to the absence of annotated datasets with explicit characteristics. Despite this limitation, it's crucial to emphasize that our explored features are subtle and interconnected. For example, subtle details like hairstyles (e.g., bangs) are strongly linked to gender, and characteristics like baldness correlate with physical appearance, also associated with gender. Additionally, attributes such as hats and eyeglasses are tied to accessories. Though not the primary focus for unlearning, these features hold potential utility for such purposes.

In future, we aspire to provide rigorous theoretical guarantees to such methods making them more dependable and safe for deployment. Moreover, the approach proposed in this work is inherently generic and can be applied to any model exhibiting parameter-space semantics. This opens up the possibility of extending such techniques to more powerful generative models, such as Diffusion models and Flow-based models. However, applying this method to iterative models like diffusion models would require a thorough investigation of their parameter-space semantics. Additionally, developing an effective strategy for negative adaptation in these models remains an open research challenge, unlike GANs, where few-shot generative adaptation is relatively

well-studied. Nonetheless, this represents a promising avenue for modern generative models, and we leave this exploration for future research.

**Broader Impact Statement**

Machine unlearning has emerged as a crucial tool for addressing privacy concerns and mitigating harmful biases in AI systems. Our method advances this field by enabling selective feature removal from pre-trained GANs without requiring access to the original training data. This capability is particularly valuable for correcting deployed models that may generate problematic or biased content, making AI systems more ethical and socially responsible. While our approach currently focuses on GANs, its success demonstrates the potential for developing similar techniques for other generative models, contributing to the broader goal of creating AI systems that can be refined and corrected post-deployment to better serve society's needs.

**Acknowledgements**

This work was supported (in part for setting up the GPU compute) by the Indian Institute of Science through a start-up grant. Piyush is supported by Government of India via Prime Minister's Research Fellowship. Atri contributed to this work as part of Uplink: IKDD Research Internship Program. Subhodip is supported by MOE Fellowship. Prathosh is supported by Infosys Foundation Young investigator award.

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

# A    Training Details

Here, we provide the details pertaining to the proposed method. Specifically, we provide the details of the pre-trained GANs and pre-trained Classifiers used in the proposed method. We also provide details pertaining to the training strategy used during Unlearning. All the experiments are performed on RTX-A6000 GPUs with 48GB memory. Our code and implementation is available at: `https://github.com/atriguha/Adapt_Unlearn`.

## A.1    Details of Pre-trained GAN

As mentioned in the main text, we use the famous StyleGAN2 architecture to obtain the pre-trained GAN. We use the open-source pytorch repository[2] for implementation. We resize the MNIST images to $32 \times 32$ and CelebA-HQ images to $256 \times 256$ to fit in the StyleGAN2 architecture. The latent space dimension for MNIST and CelebA-HQ is consequently set to $128 \times 1$ and $512 \times 1$. We train the GAN using the non-saturating adversarial loss along with path-regularization for training. We use default optimizers and hyperparameters as provided in the code for training. We train the GAN for $2 \times 10^5$ and $3.6 \times 10^5$ epochs for MNIST and CelebA-HQ respectively.

## A.2    Details of Pre-trained Classifiers

We use pre-trained classifiers to simulate the process of obtaining the feedback. More specifically, the feedbacks (positive and negative samples) are obtained by passing the generated samples (from the pre-trained GAN) through these pre-trained classifiers. The classifier classifies the generated samples into positive and negative samples. Furthermore, the classifiers are also employed for obtaining the evaluation metrics as discussed in Section-4.3 of the main text.

**MNIST**: We use simple LeNet model (LeCun et al., 1998) for classification among different digits of MNIST dataset[3]. The model is trained with a batch-size of 256 using Adam optimizer with a learning rate of $2 \times 10^{-3}$, $\beta_1 = 0.9$ and $\beta_2 = 0.999$. The model is trained for a resolution of $32 \times 32$ same as the pre-trained GAN for 12 epochs. After training the classifier has an accuracy of 99.07% on the test split of MNIST dataset.

**CelebA-HQ**: We use ResNext50 model (Xie et al., 2017) for classification among different facial attributes contained in CelebA[4]. Note that we train the classifier on normal CelebA as the ground truth values are available for it. The classifier is trained with a batch-size of 64 using Adamax optimizer with a learning rate of $2 \times 10^{-3}$, $\beta_1 = 0.9$ and $\beta_2 = 0.999$. The model is trained for a resolution of $256 \times 256$ for 10 epochs. We also employ image augmentation techniques such as horizontal flip, image resize, and cropping to improve the performance of the classifier. The trained model exhibits a test accuracy of 91.93%.

## A.3    Unlearning Hyper-parameters

Here we mention the hyper-parameters pertaining to the proposed negative adaptation and unlearning stages. As mentioned, we use an EWC regularizer during adaptation to avoid overfitting. The value of $\lambda$ (Eq.3) is set to $5 \times 10^8$ for all the experiments. Further, $\gamma$ (Eq.5) is chosen between 0.1, 1 and 10 when $\mathcal{L}_{repulsion}^{\text{IL2}}$ and $\mathcal{L}_{repulsion}^{\text{NL2}}$ are chosen as repulsion loss. It is varied between 10 and 500 when $\mathcal{L}_{repulsion}^{\text{EL2}}$ is chosen as repulsion loss. Further, the value of $\alpha$ for $\mathcal{L}_{repulsion}^{\text{EL2}}$ (Eq.7) is varied between 0.1 and 0.001. These values are chosen and adjusted to ensure that both the loss components $\mathcal{L}_{adv}^{'}$ and $\mathcal{L}_{repulsion}$ are minimized properly.

## A.4    Details of Baselines

Here, we present the details pertaining to the baselines presented in Table 1, 2, 3.

---

[2]https://github.com/rosinality/stylegan2-pytorch
[3]https://github.com/csinva/gan-vae-pretrained-pytorch/tree/master/mnist_classifier
[4]https://github.com/rgkannan676/Recognition-and-Classification-of-Facial-Attributes/

EU-$k$ and CF-$k$ (Goel et al., 2022) propose to train just the last $k$ layers of the model from scratch (in EU-$k$) or from pre-trained initialization (in CF-$k$) for unlearning on the positive samples. We employ the same strategy to GANs directly with $k = 10$ layers. Further, $\ell$1-sparse (Jia et al., 2023) proposes to use sparse weights for fine-tuning to unlearn the undesired features. To this end, they propose to use $\ell$-1 regularization while fine-tuning. Hence, for our case, we fine-tune the model on positive samples by adding an $\ell$-1 regularization on weights of the network.

For the few-shot adaptation baselines, we directly employ the provided open-source codebase of CDC[5] and RSSA[6] for adaptation on positive samples to obtain results.

## B   MNIST Qualitative Results

We present visual illustration of images generated after unlearning using various methods in figure 4. Here, we unlearn class of digits 1, 4, and 8. We observe that all the proposed methods effectively unlearn the undesired classes. Moreover, it can be seen that although extrapolation leads to unlearning, it does so at the expense of the quality of the generated images. In contrast, the quality of the generated images after unlearning using the proposed method leads to unlearning with plausible image quality. We refer the reader the reader to main text for quantitative evaluation.

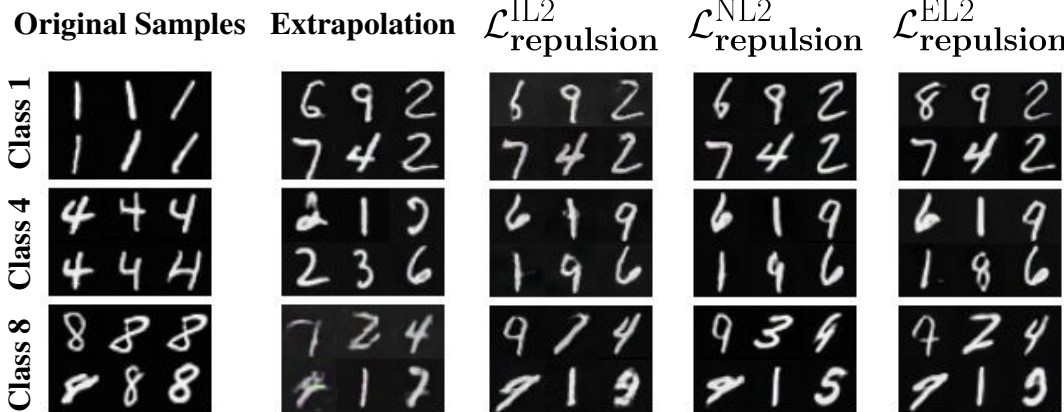

Figure 4: Results of Unlearning undesired feature (class) via different methods. The undesired class contains digits 1(top row), 4(second row), 8 (bottom row).

## C   Proofs

*Proof of Theorem 1.* The said objective function is given by-

$$\min_{\theta_P} \max_{\phi} \underset{\mathbf{x} \sim p_{G \setminus N}^{\mathcal{X}}}{\mathbb{E}} \left[\log D_\phi(\mathbf{x})\right] + \underset{\substack{\mathbf{z} \sim p_Z \\ \theta \sim p_U^\Theta}}{\mathbb{E}} \left[\log(1 - D_\phi(G_\theta(\mathbf{z})))\right] + \mathcal{L}_{repulsion}$$

$$\equiv \min_{\theta_P} \max_{\phi} \underset{\mathbf{x} \sim p_{G \setminus N}^{\mathcal{X}}}{\mathbb{E}} \left[\log D_\phi(\mathbf{x})\right] + \underset{\mathbf{x} \sim p_U^{\mathcal{X}}}{\mathbb{E}} \left[\log(1 - D_\phi(\mathbf{x}))\right] + \mathcal{L}_{repulsion} \qquad (9)$$

Since $\phi$ depends only on the first two terms of Eq. 9, the optimal discriminator as obtained in Goodfellow et al. (2014) is given as $D_{\phi^*} = \frac{p_{G \setminus N}^{\mathcal{X}}}{p_{G \setminus N}^{\mathcal{X}} + p_U^{\mathcal{X}}}$. Substituting this in Eq. 9 and using standard results from Goodfellow et al. (2014) gives -

$$\min_{\theta_P} D_{JSD}(p_{G \setminus N}^{\mathcal{X}} || p_U^{\mathcal{X}}) + \mathcal{L}_{repulsion} \qquad (10)$$

---

[5]https://github.com/utkarshojha/few-shot-gan-adaptation
[6]https://github.com/StevenShaw1999/RSSA

$\left(\mathcal{L}_{repulsion} = \mathcal{L}_{repulsion}^{\mathrm{IL2}}\right)$: Since $p_N^\Theta(\theta) = \frac{1}{|2\pi\Sigma|^{d/2}}\exp\left[\frac{1}{2}(\theta-\theta_N)^T\Sigma^{-1}(\theta-\theta_N)\right]$ and $p_U^\Theta(\theta) = \frac{1}{|2\pi\Sigma|^{d/2}}\exp\left[\frac{1}{2}(\theta-\theta_P)^T\Sigma^{-1}(\theta-\theta_P)\right]$ are both Gaussian distributions, then $D_{KL}(p_U^\Theta||p_N^\Theta) = \|\theta_P-\theta_N\|^2 \implies [D_{KL}(p_U^\Theta||p_N^\Theta)]^{-1} = \frac{1}{\|\theta_P-\theta_N\|^2} = \mathcal{L}_{repulsion}^{\mathrm{IL2}}$. Substituting in the above we get -

$$\min_{\theta_P} D_{JSD}(p_{G\setminus N}^{\mathcal{X}}||p_U^{\mathcal{X}}) + [D_{KL}(p_U^\Theta||p_N^\Theta)]^{-1} \tag{11}$$

$\left(\mathcal{L}_{repulsion} = \mathcal{L}_{repulsion}^{\mathrm{NL2}}\right)$: Similar to above argument, $D_{KL}(p_U^\Theta||p_N^\Theta) = \|\theta_P-\theta_N\|^2 \implies -D_{KL}(p_U^\Theta||p_N^\Theta) = -\|\theta_P-\theta_N\|^2 = \mathcal{L}_{repulsion}^{\mathrm{NL2}}$. Substituting in the above we get -

$$\min_{\theta_P} D_{JSD}(p_{G\setminus N}^{\mathcal{X}}||p_U^{\mathcal{X}}) - D_{KL}(p_U^\Theta||p_N^\Theta) \tag{12}$$

$\left(\mathcal{L}_{repulsion} = \mathcal{L}_{repulsion}^{\mathrm{EL2}}\right)$: Again, since $p_N^\Theta$ and $p_U^\Theta$ follow Gaussian distribution, the Hellinger divergence between two Gaussian distribution is a shifted negative Manhabolis distance between the means of the two distributions, i.e., $D_H(p_U^\Theta||p_N^\Theta) = 1 - \exp\left(-\|\theta_P-\theta_N\|^2\right) \implies 1 - D_H(p_U^\Theta||p_N^\Theta) = \exp\left(-\|\theta_P-\theta_N\|^2\right) = \mathcal{L}_{repulsion}^{\mathrm{EL2}}$. Substituting in above we get -

$$\min_{\theta_P} D_{JSD}(p_{G\setminus N}^{\mathcal{X}}||p_U^{\mathcal{X}}) - D_H(p_U^\Theta||p_N^\Theta) + 1 \tag{13}$$

$$\equiv \min_{\theta_P} D_{JSD}(p_{G\setminus N}^{\mathcal{X}}||p_U^{\mathcal{X}}) - D_H(p_U^\Theta||p_N^\Theta) \tag{14}$$

This completes the proof of all the three statements. □

*Proof of Claim 1.* For a given latent vector, $\theta \to G_\theta(z)$ is a map from parameter space to generated sample in the data space. Hence, we can use data-processing inequality to obtain -

$$D_f(p_U^{\mathcal{X}}||p_N^{\mathcal{X}}) \le D_f(p_U^\Theta||p_N^\Theta) \tag{15}$$

$$\implies D_{JSD}(p_{G\setminus N}^{\mathcal{X}}||p_U^{\mathcal{X}}) - D_f(p_U^\Theta||p_N^\Theta) \le D_{JSD}(p_{G\setminus N}^{\mathcal{X}}||p_U^{\mathcal{X}}) - D_f(p_U^{\mathcal{X}}||p_N^{\mathcal{X}}) \tag{16}$$

This completes the proof. □

# D    Results with Curated Datasets

As explained in the main paper, our method requires users to identify or annotate negative samples under feedback-based framework. This annotation is used to adapt the GAN to negative samples during the Negative Adaptation phase and subsequently retrain it on positive samples during the Unlearning phase. While human feedback is one approach to obtain these samples, curated datasets of positive and negative samples can also serve this purpose.

Curating datasets, however, can be challenging, particularly when the feature, concept, or class to be unlearned is subtle or complex and not readily available in standard datasets. In such cases, users may need to **create** a custom dataset. By contrast, our human-feedback approach involves annotating samples generated by the GAN, reducing the need for external dataset creation. Nevertheless, if a curated dataset of positive and negative samples is available, our method can be seamlessly adapted to utilize it.

To demonstrate this, we conduct experiments on all three datasets (MNIST, AFHQ, and CelebA-HQ) using curated dataset samples in both the Negative Adaptation and Unlearning phases. Specifically, for the Negative Adaptation phase, we use samples from the original dataset that were pre-annotated with the undesired feature or label. For instance, in the CelebA-HQ dataset, we selected all samples with a positive label for the undesired feature (e.g., Bangs). The GAN is then adapted to these samples using the training objective described in Section 3.2 (Eq. 1 of the main paper) to obtain the parameters $\theta_N$. In the Unlearning phase, the GAN was retrained on the remaining dataset samples using the objective described in Section 3.3 (Eq. 4).

We performed these experiments on MNIST (unlearning 'Class 8'), AFHQ (unlearning 'Cat'), and CelebA-HQ (unlearning 'Bangs'). The results of these experiments are summarized in Table 6, comparing the performance of using curated dataset samples against GAN-generated samples.

Table 6: Comparison of results using curated dataset samples versus GAN-generated samples for negative adaptation. PUL ($\uparrow$), FID ($\downarrow$), and Ret-FID ($\downarrow$).

| Method | MNIST (Class 8) | | | AFHQ (Cat) | | | CelebA-HQ (Bangs) | | |
|---|---|---|---|---|---|---|---|---|---|
| | PUL | FID | Ret-FID | PUL | FID | Ret-FID | PUL | FID | Ret-FID |
| w/ dataset samples | $97.30 \pm 1.20$ | $7.79 \pm 0.86$ | $10.7 \pm 0.52$ | $94.26 \pm 1.08$ | $11.88 \pm 0.81$ | $6.13 \pm 1.66$ | $90.40 \pm 0.91$ | $10.04 \pm 0.49$ | $6.05 \pm 1.25$ |
| w/ GAN samples (using $\mathcal{L}_{repulsion}^{EL2}$) | $95.22 \pm 0.34$ | $8.80 \pm 0.52$ | $5.68 \pm 0.10$ | $95.76 \pm 0.25$ | $16.50 \pm 0.12$ | $8.17 \pm 0.14$ | $90.45 \pm 1.02$ | $11.16 \pm 0.08$ | $7.94 \pm 0.32$ |

As shown in Table 6, using curated dataset samples during the Negative Adaptation phase achieves performance comparable to that of GAN-generated samples. This demonstrates the flexibility of our method, allowing users to choose the most convenient approach depending on the availability of datasets. These results will be included in the supplementary material to provide further clarity on this point.

# E   Effect on other features after Unlearning

As previously discussed, the unlearning procedure aims to exclusively erase undesired features without impacting other features. Consequently, it becomes imperative to assess whether the unlearning process exerts any influence on other features. To address this concern, we introduce plots illustrating the percentage change in the presence of other features.

Specifically, we undertake the generation of $15,000$ random samples from both the pre-trained GAN and the GAN after the unlearning procedure. Subsequently, employing the pre-trained classifiers (for comprehensive details, please refer to the Supplementary), we calculate the occurrence of specific features within the two GANs. We report the percentage change in these numbers to demonstrate how has the unlearning process affected this feature. Hence, a lower percentage change is better as it means that the other features are not affected after unlearning. This experiment is repeated across multiple features, and our findings after unlearning Bangs in CelebA-HQ are depicted in figure 5. We observe that for all the features, unlearning via extrapolation leads to significant changes in other features. Whereas, unlearning via the proposed method leads to minor changes in the features. For instance, since Bangs are highly correlated with gender, we observe that unlearning Bangs via extrapolation leads to an increase in Males. However, unlearning via the proposed method leads to minor changes in the number of samples with Male features. This shows that the proposed method is effective in erasing the undesired feature while preserving other features. It can be observed that in majority of the cases, extrapolation leads to significant change in the features, indicating that unlearning via extrapolation leads to significant change in other features as well. This also indicates that extrapolation leads to unlearning of several correlated features. On the other hand, we observe that unlearning using the proposed method with $\mathcal{L}_{repulsion}^{\text{EL2}}$ gives the least change in most of the cases. This illustrates the efficacy of the proposed method in preserving features other than the unlearnt feature.

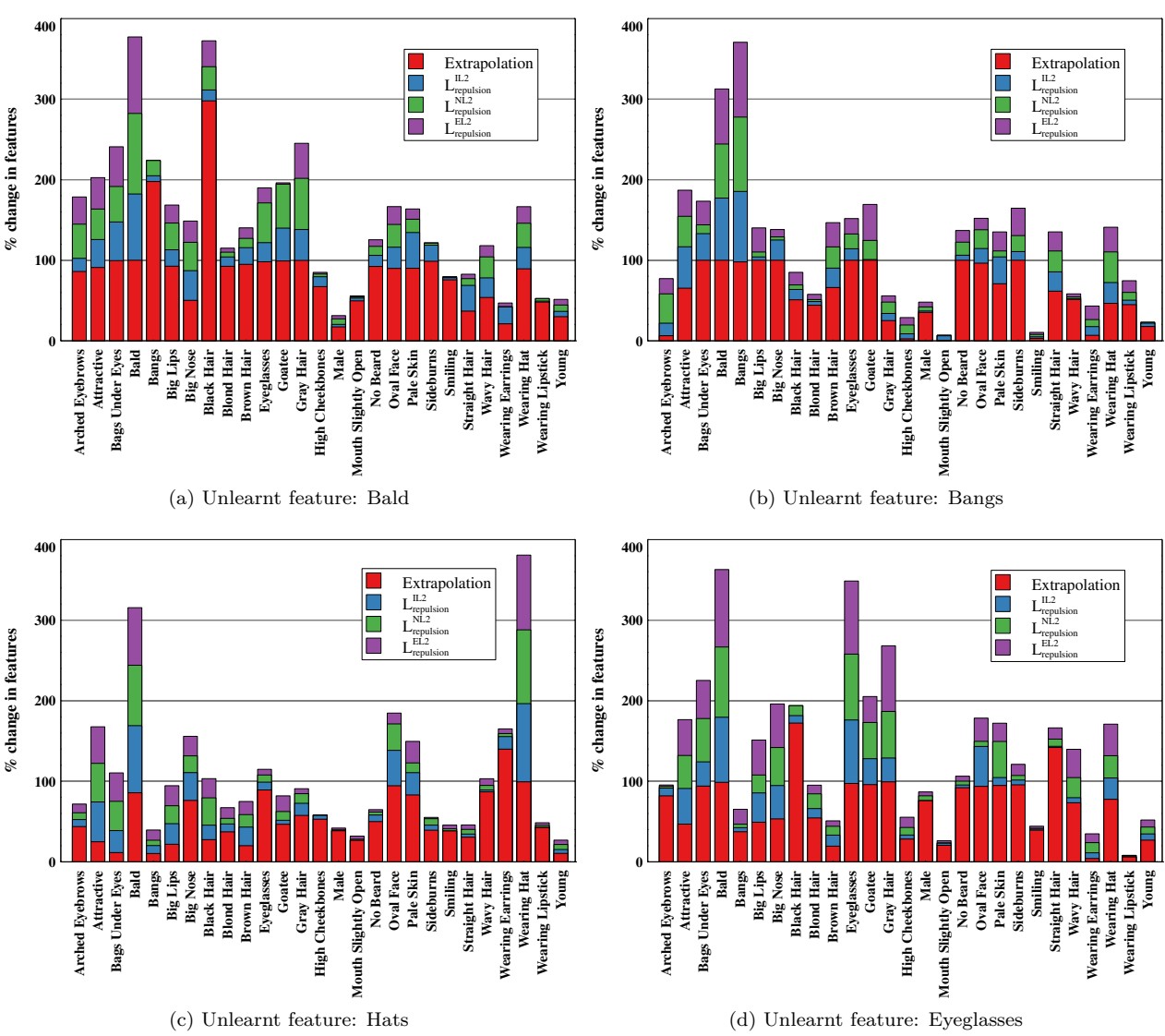

Figure 5: Percentage (%) change in several features after unlearning specific features. A lower percentage change denotes that the method has successfully preserved that feature after unlearning. Here, we clip the bar to 100% if the percentage change in that feature is more than 100% to make the plots legible.

## F  Visualization of Generated samples

Here, we provide visualization of random samples generated from the original GAN and the GAN after unlearning to give an intuitive/qualitative idea of generation quality. We provide the results by using $\mathcal{L}_{repulsion}^{EL2}$ as it gives the best performance in most of the cases. These results are presented for all the datasets in figure 6, 7, 8. While the results in Section 5.4 provide quantitative idea of generation quality, these visualizations provide qualitative idea of the same.

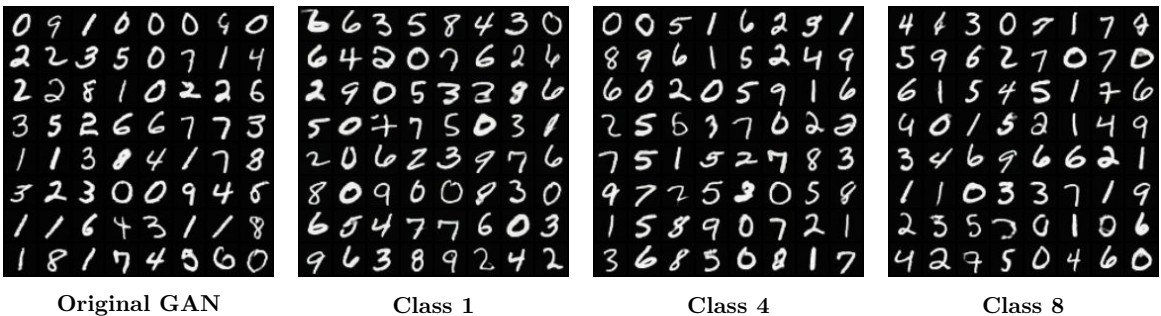

| Original GAN | Class 1 | Class 4 | Class 8 |

Figure 6: Visualization of random MNIST samples generated before and after unlearning.

**Original GAN**

**Cat**

**Dog**

**Wild**

Figure 7: Visualization of random AFHQ samples generated before and after unlearning.

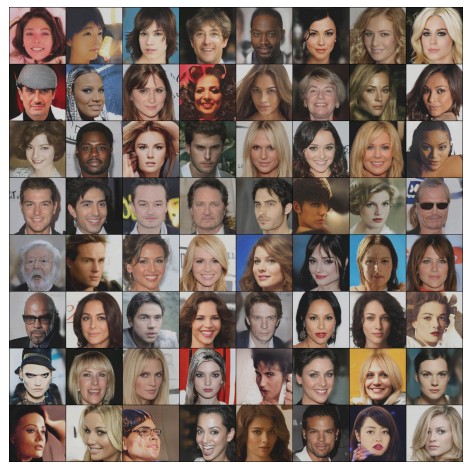

**Original GAN**

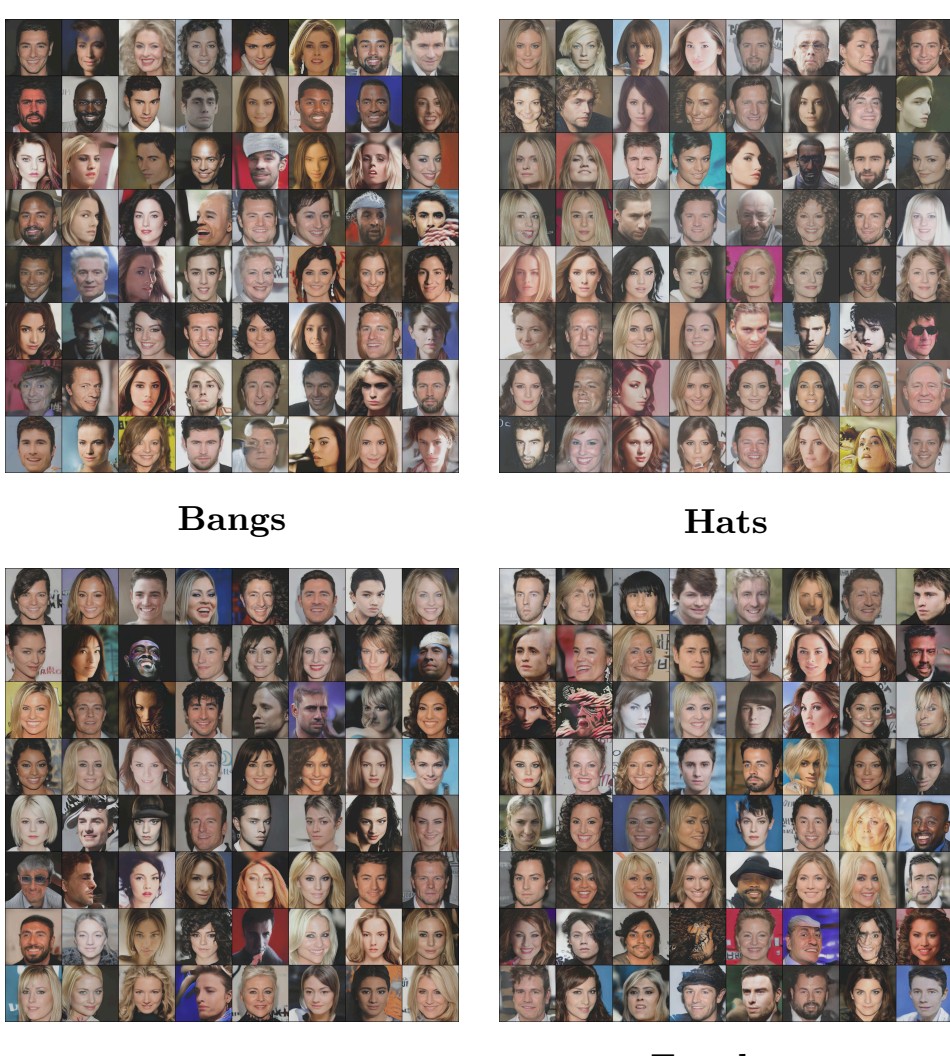

**Bangs**                    **Hats**

**Bald head**                **Eyeglass**

Figure 8: Visualization of random CelebA-HQ samples generated before and after unlearning.

