# OpenReview forum: "Adapt then Unlearn: Exploring Parameter Space Semantics for Unlearning in Generative Adversarial Networks"
_TMLR — Accepted by TMLR_

### Review · Reviewer_ouU3 · 2024-08-19

**Summary Of Contributions:**

The paper proposes a new ¨unlearning¨ technique, i.e., how to remove the given mode a distribution, i.e. how to re;ove one of the Gaussian in a mixture of Gaussian. This problem has obvious applications to prevent models from generating harmful content.

**Audience:**

Yes

**Broader Impact Concerns:**

No concerns

**Claims And Evidence:**

Yes

**Requested Changes:**

Clarify the proposed technique, maybe by writing the equations more exhaustively

**Strengths And Weaknesses:**

Strength:
The problem is hard and very interesting, and the experimental results are convincing.

Weaknesses:
IMO the paper has two main weaknesses:
- 1 applicability of the proposed method, from what I understand, the method requires human feedback to annotate examples to remove, which can be tedious and costly (this is however standard in LLMs with RLHF). Maybe a more realistic/interesting would be to suppose that one has access to a dataset of the mode to remove (e.g only gods, or only harmful content), this would be another project, but out of curiosity would it be possible to adapt the proposed technique ti this setting?
- 2 clarify. I am not an expert in unlearning, but I am not sure I fully understand the method, here is a list of clarification questions

- why is $\mathcal{L}_{adv}$ named thi was? is this an adversarial term?
- From what I understand, the repulsion is done in the parameters space?
- In Equation 3, could you clarify what does $\mathcal{L}(\mathcal{S}_n | \theta_G)$ mean? It seems that you compare samples against parameters.
- the technique seems quite generic, why make it specific to GANs? Could it be applied to diffusion/flow matching as well?


Minor:
- ¨fischer' >> ¨Fisher¨
- ¨Francesco D’Angelo and Vincent Fortuin¨ is duplicated in the bibliography
- In Thm 1, why not directly replace $\Sigma$ by $I$?

---

> ### Author Response · Authors · 2024-08-27
> **Author Response for Comments by Reviewer ouU3**
>
> We thank the reviewer for their feedback and clarifications, which have greatly contributed to improving the paper. We've addressed all the comments to the best of our ability. However, please let us know if any further clarification is needed.
>
> 1. Applicability of the Proposed method:
>
>     - We appreciate your comment. As you correctly pointed out, our method requires users to identify or annotate negative samples (those with undesired features), similar to RLHF in LLMs.
>     - The purpose of this annotation is to adapt the GAN on negative samples (in Negative adaptation phase) and then re-train the GAN on positive samples (in Unlearning phase). Human feedback is simply one way to obtain these samples.
>     - These samples can be acquired through various methods. As demonstrated in the paper, one approach is via human feedback. Alternatively, a curated dataset (for negative and positive samples) can also serve this purpose.
>     - However, curating such datasets can be challenging. Specifically, when the feature/concept/class that is to be unlearnt is subtle/complex in that, they might not be annotated/available in any of the standard datasets. In such cases, users might need to **create** such a dataset themselves.
>     - In contrast, our current human-feedback approach involves annotating readily available samples generated by the GAN.
>     - Nevertheless, if a dataset for positive and negative samples is readily available, our method can be easily adapted. The GAN can be trained on this dataset to obtain the parameters $\theta_N$, which can then be used in the Unlearning phase.
>     - To demonstrate this, we conduct experiments across all three datasets with different undesired features. Specifically, instead of using GAN-generated samples in both the phases, we used samples from the original dataset to obtain $\theta_N$ as follows:
>         - We take samples from the original dataset which are pre-annotated with the undesired label. For instance, in CelebA-HQ, we take all the samples which have positive label for the undesired feature (e.g. Bangs).
>         - Then, we adapt the original GAN on these samples as explained in Section 3.2 using Eq. 1 as the training objective to obtain $\theta_N$.
>         - Finally, we unlearn these features by re-training the GAN on the remaining samples from the dataset as explained in Section 3.3 using Eq. 4 as the training objective.
>
>     - We conduct these experiments on all the three datasets: MNIST (unlearning 'Class 8'), AFHQ (unlearning 'Cat') and CelebA-HQ (unlearning 'Bangs'). The results of these experiments are presented in the following table:
>
> |                                                        | MNIST (Class 8) |                 |                 |    AFHQ (Cat)    |                  |                 | CelebA-HQ (Bangs) |                  |                 |
> |--------------------------------------------------------|:---------------:|:---------------:|:---------------:|:----------------:|:----------------:|:---------------:|:-----------------:|:----------------:|:---------------:|
> |                                                        |       PUL       |       FID       |     Ret-FID     |        PUL       |        FID       |     Ret-FID     |        PUL        |        FID       |     Ret-FID     |
> | w/ dataset samples                                     | $97.30\pm 1.20$ | $7.79 \pm 0.86$ | $10.7 \pm 0.52$ | $94.26 \pm 1.08$ | $11.88 \pm 0.81$ | $6.13 \pm 1.66$ | $90.40 \pm 0.91$  | $10.04 \pm 0.49$ | $6.05 \pm 1.25$ |
> | w/ GAN samples (using $\mathcal{L}_{repulsion}^{EL2}$) | $95.22\pm 0.34$ | $8.80 \pm 0.52$ | $5.68 \pm 0.10$ | $95.76 \pm 0.25$ | $16.50 \pm 0.12$ | $8.17 \pm 0.14$ | $90.45 \pm 1.02$  | $11.16 \pm 0.08$ | $7.94 \pm 0.32$ |
>
> - We observe that using the original dataset for the negative adaptation phase yields performance comparable to that obtained with GAN-generated samples. Therefore, either approach can be employed based on convenience and the availability of such a dataset. We will include these results in a new section in the appendix and provide further clarification on this point.

---

> ### Author Response · Authors · 2024-08-27
>
> 2. *Why is $\mathcal{L}_{adv}$ named this way? is this an adversarial term?*
>
>     Thank you for this question. Yes, $\mathcal{L}_{adv}$ is indeed an adversarial term. There are two adversarial terms used across the two phases:
>
>     (a) *$\mathcal{L}_{adv}$ in the Negative Adaptation Phase* (Eq. 1): In this first phase, we adapt the original GAN to the negative samples (those identified as undesired through user feedback). We do this by adapting the GAN on these negative samples using an adversarial loss, where the negative samples are treated as real. This phase also includes a regularization term, $\mathcal{L}_{adapt}$, to prevent catastrophic forgetting. The resulting parameters from this phase are denoted as $\theta_N$.
>
>     (b) *$\mathcal{L}_{adv}^{'}$ in the Unlearning Phase* (Eq. 4): In the second phase, we retrain the original GAN on positive samples (those **not** marked as undesired) using an adversarial loss, where the positive samples are treated as real. Additionally, a repulsion loss is applied to move the parameters away from $\theta_N$ obtained in the Negative Adaptation phase.
>
> 3. *..the repulsion is done in the parameters space?*
>
>     Yes, the repulsion loss is employed in the parameter space, where the learnt generator parameter ($\theta_P$) is encouraged to move away from $\theta_N$ to supress the undesired features.
>
> 4. *In Equation 3, could you clarify what does $\mathcal{L}(S_n|\theta_G)$ mean? It seems that you compare samples against parameters.*
>
>     Apologies for the confusion. $\mathcal{L}(S_n|\theta_G)$ refers to the log-likelihood function for the samples $S_n$ generated by the GAN with parameters $\theta_G$. Specifically, it represents $\log p_{\theta_G}(S_n)$, which is the log-likelihood of the negative samples under the generator's distribution with parameter $\theta_G$. This notation is directly borrowed from [1], where EWC was first introduced for few-shot generative adaptation (see Equation 2 in their paper). This term can be estimated by calculating the binary cross-entropy of the discriminator's output, as shown in [1]. We will clarify this in the revised manuscript to prevent any misunderstandings.
>
> 5. *..the technique seems quite generic, why make it specific to GANs? Could it be applied to diffusion/flow matching as well?*
>
>     Thank you for this insightful comment. The technique is indeed conceptually generic and could potentially be extended to modern models like diffusion or flow matching—an exciting direction for future research. However, we note that applying it to iterative models like diffusion models would require an in-depth study of the parameter space semantics. Additionally, developing an effective method for negative adaptation in these models is still an active research area, unlike in GANs, where few-shot generative adaptation is well-studied. Nonetheless, this is a promising direction for modern generative models, and we will discuss it further in the Future Works section.
>
> 6. Minor Comments:
>
>     (i) *¨fischer' >> ¨Fisher¨*: Thanks for pointing out, this will be corrected.
>
>     (ii) *¨Francesco D’Angelo and Vincent Fortuin¨ is duplicated in the bibliography*: Thanks for noticing, will be rectified in revision.
>
>     (iii) *In Thm 1, why not directly replace $\Sigma$ by $I$?*: Yes, this can be done, we wrote the Theorem in full generality. However, we will simplify this to avoid any confusion and for the ease of reader.
>
>
>
> [1] Li, Yijun, et al. "Few-shot image generation with elastic weight consolidation." 34th Conference on Neural Information Processing Systems (NeurIPS 2020).

---

> > ### Author Response · Authors · 2024-11-28
> > **Revised Manuscript**
> >
> > We again thank you for your time and effort to review our paper. We have revised our manuscript to accomodate and address your suggestions and concerns.
> >
> > 1. As suggested, additional text has been added to the end of Section 3.3 to emphasize the generality of the proposed method.
> > 2. A discussion on how the method can potentially be extended to modern generative models like Diffusion models and Flow-based models has been included in Future Works.
> > 3. Section 3.3 now explicitly distinguishes between the two adversarial terms.
> > 4. The notation of the log-likelihood function in Section 3.2 has been updated, as requested, to avoid confusion. We have also clarified how this term is calculated using the discriminator's output.
> > 5. Section 3.4 now explicitly states that the repulsion term operates in parameter space.
> > 6. Results for unlearning with pre-curated datasets for negative and positive samples have been added in Supplementary Section D, as suggested.
> >
> > We sincerely hope these changes adequately address your concerns and suggestions. We are happy to make further adjustments based on any additional feedback.

---

### Review · Reviewer_H1FD · 2024-09-16

**Summary Of Contributions:**

This paper introduces a new method for unlearning generative models so they do not generate images
from a certain class or with certain features. The method is demonstrated to work on StyleGAN2 and
works in the few shot setting.

**Audience:**

Yes

**Broader Impact Concerns:**

I have no ethical concerns about this work that would require adding an ethical impact statement.

**Claims And Evidence:**

Yes

**Requested Changes:**

It would be helpful to better explain how this work differs from Sun et al. in the related work section. It is already compared against in the experimental section.

The paper would be strengthened if it could be compared against other methods for feature unlearning. It seems all comparisons are only for class unlearning.

**Strengths And Weaknesses:**

Strengths

The paper is very well-motivated, organised, and written. The experimental section is throrough well-thought out and
includes detailed ablation studies that motivate the contribution of the method. It is compared to the relevant
related work where it makes sense. The theoretical contributions offer interesting connections to other losses in
the GAN literature.

Weaknesses

The main challenge for the paper is that the contribution is not novel. Sun et al. 2023 also
introduce a cascade based unlearning method which uses StyleGAN2. The difference seems to be
more that this paper uses one of several revulsion losses while Sun et al. use a different
loss entirely. It would be better to highlight this difference as it seems both papers use
the same GAN architecture.

---

> ### Author Response · Authors · 2024-09-24
> **Author Response for Comments by Reviewer H1FD**
>
> We sincerely thank the reviewer for their valuable comments and insights. These have been instrumental in improving the clarity and quality of the manuscript. We hope our responses sufficiently address the reviewer’s questions and concerns. However, we are more than happy to engage in further discussion if any additional questions remain.
>
> 1. **Novelty and Differences from Sun et. al.**
>
> We thank the Reviewer for the insightful comments.
> While both the work by Sun et al. and ours are similar in that both operate on StyleGAN2 architecture, there are several significant differences, which are outlined below:
>
> -  *Unlearning Setting*:
>         There is a fundamental difference in the unlearning setting between Sun et al. and our method. To reduce the generation of undesired samples, Sun et. al. proposes to forget undesired samples from the training dataset. Specifically, they assume access to samples from the training dataset. This is somewhat restrictive since users typically don’t have access to the training data [1,2]. \
>         In contrast, our approach operates in a more practical setting where samples corresponding to unlearning are obtained from the user feedback, making it more applicable in real-world scenarios.
>
> - *Methodology*:
>         In terms of methodology, Sun et al. propose to *patch* the latent space of the GAN with representative samples. They suggest various strategies for generating these representative samples, such as using 'average samples' or 'other class samples' (cf. Section 4.3 of their paper). However, imposing such constraints on the latent space may lead to suboptimal latent-space semantics, potentially harming the quality of generated images. To address this, we avoid manipulating the latent space directly. Instead, we focus on parameter-space semantics, where we identify generator parameters that produce undesired samples (Stage-1: Negative Adaptation), and then retrain the GAN to avoid these parameters (Stage-2: Unlearning Phase).\
>        Additionally, since the latent space naturally adjusts based on changes in the parameter space (as shown in Fig. 1(b) of our manuscript), we find it sufficient to focus on parameter-space semantics alone, as it automatically handles latent-space semantics as well.\
>         To our knowledge, these insights into parameter-space semantics have not been explored in the context of unlearning, making our approach novel.
>
> - *Theoretical Guarantees*:
>         Finally, as you mentioned, our method offers theoretical guarantees, which are absent in Sun et al.’s approach. Theoretical guarantees are crucial in areas concerning privacy and fairness, which makes our method an important step forward in studying unlearning in generative models. We believe this aspect holds significant potential for further research.
>
>
>
> 2. **Comparisons for Feature Unlearning**
>
>     - We apologize for the confusion. In our experiments, we work with both feature-level unlearning and class-level unlearning. The key distinction is that in feature-level unlearning, an image can possess multiple features simultaneously, while in class-level unlearning, an image from one class cannot belong to any other class. Put differently, if we treat each feature as a class, in feature-level unlearning, an image can belong to multiple classes, whereas in class-level unlearning, an image can only belong to one class. For example, an image of a person with bangs can be male or female, but an image labeled as the digit 'one' cannot have any other class label.
>     - In our experiments, we used the CelebA-HQ dataset for feature-level unlearning, where we unlearn features like 'bangs' or 'baldness', etc. For class-level unlearning, we used the MNIST and AFHQ datasets.
>     - We will expand on this distinction in the revised manuscript to ensure clarity.
>
>
> [1] Chundawat, Vikram S., et al. "Zero-shot machine unlearning." IEEE Transactions on Information Forensics and Security 18 (2023): 2345-2354.\
> [2] Heng, Alvin, and Harold Soh. "Selective amnesia: A continual learning approach to forgetting in deep generative models." Advances in Neural Information Processing Systems 36 (2024).

---

> > ### Author Response · Authors · 2024-11-28
> > **Revised Manuscript**
> >
> > We again thank you for your time and effort to review our paper. We have revised our manuscript to accomodate and address your suggestions and concerns.
> >
> > 1. Additional text has been added in Section 2.1 to clarify the distinction between our method and Cascaded Unlearning proposed in Sun et al.
> > 2. Text has been added in Section 5.1 to clearly explain the difference between feature unlearning and class unlearning, as requested. We have also highlighted that MNIST and AFHQ are used for demonstrating class unlearning, while CelebA-HQ is used for feature unlearning.
> >
> > We sincerely hope these changes adequately address your concerns and suggestions. We are happy to make further adjustments based on any additional feedback.

---

### Review · Reviewer_cEyE · 2024-11-23

**Summary Of Contributions:**

The authors’ goal is “unlearning” in generative models, specifically GANs. This is not the usual idea of “unlearning”, meaning removing any trace of certain training data, but rather suppressing outputs with specific undesired features without overly distorting the other outputs.

In their model, a user will obtain some samples that show the undesired feature. These are used to adapt the base GAN using an off-the-shelf few-shot adaptation technique. The result is a GAN which tends to output negative features. Then, the base GAN is trained again, this time on positive samples, but with an additional regularization term to steer its parameters away from the negative-adapted GAN. Different choices of regularization term have slightly different interpretations in terms of which divergence the GAN training procedure is approximating.

The authors use StyleGAN2 on several standard datasets (MNIST, AFHQ, and CelebA-HQ). In these experiments, they train a simple classifier to act as a simulated human (this seems like a reasonable choice and doesn’t undermine their method).

In the experiments, their method works substantially better than baselines for unlearning, although at the cost of some quality as measured by FID. They also have an ablation study giving some evidence that their repulsion loss is important to performance.

**Audience:**

Yes

**Broader Impact Concerns:**

No broader impact concerns.

**Claims And Evidence:**

Yes

**Requested Changes:**

The table of results is very tiny and hard to read. I would suggest that the authors split it up somehow to allow for bigger font. Fortunately, TMLR doesn't have page limits like CS conferences.

**Strengths And Weaknesses:**

Strengths:

- the method does seem to work well at its intended goal. The experiments reasonably convinced me of this.

Weaknesses:

- it seems to just be a bunch of existing methods glued together. however, if it works, I guess one shouldn't count this against the technique.

- I didn't find any of the theoretical results to be very enlightening, although I didn't catch any errors.

- I have no intuitive sense of "how bad" the decline in FID due to their method might be, in terms of actual perceived image quality.

---

> ### Author Response · Authors · 2024-11-27
> **Response to Reviewer cEyE**
>
> Thank you for your valuable comments. We sincerely hope our responses address your questions and concerns. We would be more than happy to engage in further discussion.
>
> > 1. It seems to just be a bunch of existing methods glued together. However, if it works, I guess one shouldn't count this against the technique.
>
> Thanks for this comment.
> - While we acknowledge that the concept of parameter-space semantics is not entirely new and has been explored in prior work [1], we believe that its application specifically for unlearning undesired features remains unexplored to the best of our knowledge.
> - Moreover, as shown in Fig. 1(b), employing parameter-space semantics alone is insufficient as it hampers the generation quality. This insight leads us to devise a two-stage procedure, as described in Section 3. First, we obtain the negative parameters, $\theta_N$, using a few-shot adaptation or fine-tuning approach like EWC. Subsequently, we fine-tune the original GAN on positive samples while incorporating a novel repulsion regularizer to effectively unlearn the undesired features.
> - Thus, while our approach leverages existing methodologies, each component is purposefully directed towards addressing the challenge of unlearning specific undesired features.
>
> > 2. I didn't find any of the theoretical results to be very enlightening, although I didn't catch any errors.
>
> We note that our theory plays a critical role in understanding the training dynamics of the GAN during the Unlearning phase. Specifically, the analysis highlights two key aspects:
>
> (a) The minimization of the JS-divergence between the learned data distribution and the implicit generator distribution after removing the support of negative samples.
>
> (b) The maximization of a divergence measure between the parameter distribution during unlearning and the parameter distribution associated with generating negative samples.
>
> While (a) is relatively straightforward, (b) provides valuable insights into the effect of the regularization term, which is particularly interesting. The regularization term ensures that the current parameter distribution moves away from the one responsible for generating undesired features, effectively aligning with the primary goal of unlearning. Furthermore, this behavior has been shown to be crucial for unlearning in Bayesian settings, as demonstrated in [2].
> We believe these theoretical results offer a meaningful contribution to the understanding of unlearning dynamics in GANs pertaining to the proposed method.
>
> > 3. I have no intuitive sense of "how bad" the decline in FID due to their method might be, in terms of actual perceived image quality.
>
> - Thank you for this comment. We appreciate your concern about understanding the impact of FID decline on perceived image quality. While we reported the decline in FID on page 11 of the manuscript, we realize that we did not provide the original FID values for comparison, and we apologize for this oversight. The original FID scores for the pre-trained GANs on different datasets are as follows: MNIST: 5.40, AFHQ: 8.10, and CelebA-HQ: 5.30. We will include these baseline FID values in the revised manuscript for better clarity.
> - Additionally, to provide a more intuitive understanding, we will add visualizations of images generated by the pre-trained GAN and the GAN after unlearning in the supplementary material. This should help illustrate the impact of unlearning on image quality.
>
> > The table of results is very tiny and hard to read. I would suggest that the authors split it up somehow to allow for bigger font.
>
> Thank you for this helpful suggestion. We will split the table into three separate tables, one for each dataset. This will allow us to increase the font size and improve overall readability.
>
>
>
> [1] Cherepkov, Anton, Andrey Voynov, and Artem Babenko. "Navigating the GAN parameter space for semantic image editing." Proceedings of the IEEE/CVF conference on computer vision and pattern recognition. 2021.
>
> [2] Nguyen, Quoc Phong, Bryan Kian Hsiang Low, and Patrick Jaillet. "Variational bayesian unlearning." Advances in Neural Information Processing Systems 33 (2020).

---

> > ### Author Response · Authors · 2024-11-28
> > **Revised Manuscript**
> >
> > We again thank you for your time and effort to review our paper. We have revised our manuscript to accomodate and address your suggestions and concerns.
> >
> > 1. The original baseline comparison table has been split into three separate tables corresponding to each dataset. This change enhances readability and visibility, as suggested.
> > 2. The FID of pre-trained GANs for each dataset has been explicitly mentioned, as pointed out. Additionally, Section F of the Supplementary now includes visualizations of samples generated before and after unlearning.
> > 3. Section 4 has been expanded to provide deeper insights and highlight the essence of the theoretical results.
> >
> > We sincerely hope these changes adequately address your concerns and suggestions. We are happy to make further adjustments based on any additional feedback.

---

### Author Response · Authors · 2024-11-28
**Revision Summary**

We sincerely thank all the reviewers for their valuable time and effort in reviewing our paper. Their constructive feedback has greatly helped us improve the manuscript in terms of presentation, clarity, and exploring additional aspects of our method. We have addressed the concerns raised by each reviewer in our responses and revised the manuscript accordingly. Major changes in the paper have been highlighted in 'blue' color. Below is a brief summary of the revisions made in the updated manuscript:

### 1. **Improved Presentation**
   - **(a)** The original baseline comparison table has been split into three separate tables corresponding to each dataset. This change enhances readability and visibility, as suggested by Reviewer **cEyE**.
   - **(b)** The formatting of the pseudo-code has been updated for better readability and aesthetics.

### 2. **Distinction from Sun et al.**
   - Additional text has been added in Section 2.1 to clarify the distinction between our method and Cascaded Unlearning proposed in Sun et al.

### 3. **Difference Between Feature Unlearning and Class Unlearning**
   - Text has been added in Section 5.1 to clearly explain the difference between feature unlearning and class unlearning, as requested by Reviewer **H1FD**. We have also highlighted that MNIST and AFHQ are used for demonstrating class unlearning, while CelebA-HQ is used for feature unlearning.

### 4. **Clarification on Methodology**
   - **(a)** Section 3.3 now explicitly distinguishes between the two adversarial terms.
   - **(b)** The notation of the log-likelihood function in Section 3.2 has been updated, as requested by Reviewer **ouU3**, to avoid confusion. We have also clarified how this term is calculated using the discriminator's output.
   - **(c)** Section 3.4 now explicitly states that the repulsion term operates in parameter space.
   - **(d)** Section 4 has been expanded to provide deeper insights and highlight the essence of the theoretical results, as suggested by Reviewer **cEyE**.

### 5. **Method's Generality**
   - **(a)** As suggested by Reviewer **ouU3**, additional text has been added to the end of Section 3.3 to emphasize the generality of the proposed method. Supplementary Section D now includes results related to this aspect.
   - **(b)** A discussion on how the method can potentially be extended to modern generative models like Diffusion models and Flow-based models has been included in Future Works.

### 6. **Additional Results**
   - **(a)** The FID of pre-trained GANs for each dataset has been explicitly mentioned, as pointed out by Reviewer **cEyE**. Additionally, Section F of the Supplementary now includes visualizations of samples generated before and after unlearning.
   - **(b)** Results for unlearning with pre-curated datasets for negative and positive samples have been added in Supplementary Section D, as suggested by Reviewer **ouU3**.

We sincerely hope these changes adequately address the reviewers' concerns and suggestions. We are happy to make further adjustments based on additional feedback from the reviewers.

---

### Author Response · Authors · 2024-12-27
**A Gentle reminder on review responses.**

Dear Reviewers and AEs,

We extend our seasonal greetings to you all.

We are writing this to gentle remind you of the next course on action on our submission. We have provided a detailed response and revised the paper based on your comments.

Thank you for your kind consideration.

Authors

---

> ### Comment · Action_Editor_mddz · 2024-12-27
>
> Thank you for the note :)  I am working to move along the last steps of the review process.  Sorry about the delay!

---

> > ### Author Response · Authors · 2024-12-27
> > **Response.**
> >
> > Thanks for the quick response.
> >
> > We look forward to hearing from you soon.
> >
> > New year greetings once again.
> >
> > Authors.

---

### Decision · Action_Editor_mddz · 2025-01-08

**Recommendation:** Accept as is

**Comment:**

The reviewers believe that the paper meets the standards and criteria for acceptance to TMLR.

**Audience:**

Yes, the topic has generated a lot of interest within the TMLR community.

**Claims And Evidence:**

The reviewers generally agreed that the paper supports its claims with evidence and meets the TMLR standards.  One reviewer had concerns that the method isn't specified well enough to be implemented, so we encourage the authors to continue to revise the paper to make sure those concerns are fully addressed.